# Let's (not) just put things in Context: Test-time Training for Long-context LLMs

**Rachit Bansal**[1][2][*] **Aston Zhang**[3]
**Rishabh Tiwari**[4][*] **Lovish Madaan**[5] **Sai Surya Duvvuri**[6][*] **Devvrit Khatri**[6][*]
**David Brandfonbrener**[7] **David Alvarez-Melis**[1][2] **Praj Bhargava**[5] **Mihir Kale**[8] **Samy Jelasi**[1]

[1]Harvard University [2]Kempner Institute at Harvard [3]OpenAI [4]UC Berkeley
[5]Meta Superintelligence Labs [6]UT Austin [7]Anthropic [8]Microsoft Research

## ABSTRACT

Progress on training and architecture strategies have enabled LLMs with millions of tokens in context length. However, empirical evidence suggests that such long-context LLMs can *consume* far more text than they can reliably *use*. On the other hand, it has been shown that inference-time compute can be used to scale performance of LLMs, often by generating thinking tokens, on challenging tasks involving multi-step reasoning. Through controlled experiments on sandbox long-context tasks, we find that such inference-time strategies show rapid diminishing returns, and fail at long context. We attribute these failures to *score dilution*, a phenomenon inherent to static self-attention. Further, we show that current inference-time strategies cannot retrieve relevant long-context signals under certain conditions. We propose *query-only test-time-training* (qTTT) that, through targeted gradients updates on the given context, provably overcomes limitations of static self-attention. We find that this simple shift in how inference-time compute is spent leads to consistently large performance improvements across models and long-context benchmarks. qTTT leads to massive 12.6% and 14.1% points improvements for Qwen3-4B on average across subsets of LongBench-v2 and ZeroScrolls benchmarks. The takeaway is practical: for long context, a small amount of context-specific training is a better use of inference compute than current inference-time scaling strategies like producing more thinking tokens.

## 1 INTRODUCTION

Many ambitious LLM use-cases are rooted in long context: analyzing scientific corpora (Katz et al., 2023; Taylor et al., 2022), synthesizing books (Kryscinski et al., 2022), maintaining rich multi-turn histories (Park et al., 2023; Zhou et al., 2024), and reasoning over large multi-file code repositories (Jimenez et al., 2024; Zhang et al., 2023). Recent progress in pre-training and architectural strategies have enabled context windows with millions of tokens (Yang et al., 2025; Ding et al., 2024; Reid et al., 2024; Anthropic, 2024). In practice, however, persistent failure modes remain: models miss clauses buried in lengthy documents, overlook function definitions deep in repositories, or fail to retrieve facts from prior turns even when the relevant content is present "in context" (Liu et al., 2024; Hsieh et al., 2024; Kamradt, 2024).

Concurrently, there is a growing interest in using inference-time compute to overcome limitations of vanilla transformer models. Methods such as chain-of-thought "thinking" tokens (Wei et al., 2022b), best-of-$n$ (Nakano et al., 2021; Stiennon et al., 2020), and other "thinking" strategies (Zelikman et al., 2024) have shown promise. However, all these methods generate additional tokens with the same static attention mechanism that is already under-allocating mass to the evidence.

---

Correspondence: rachitbansal@g.harvard.edu, az@astonzhang.com
[*]Work done during an internship at Meta Superintelligence Labs.

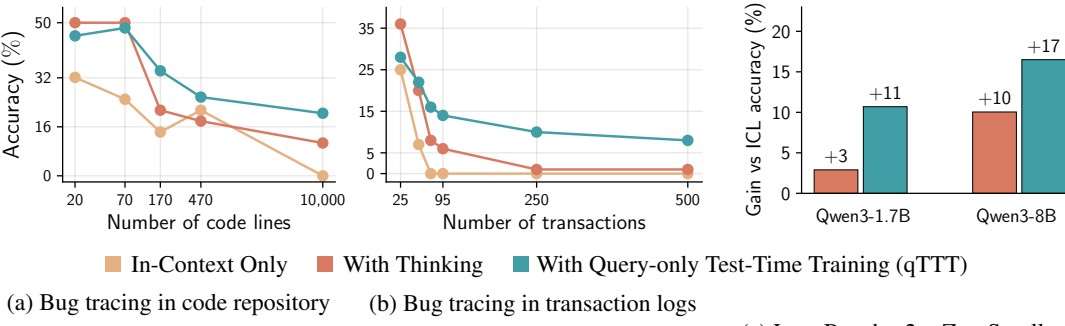

(a) Bug tracing in code repository    (b) Bug tracing in transaction logs

(c) LongBench-v2 + ZeroScrolls

Figure 1: Query-only test-time training uses inference-time compute more effectively than "thinking" tokens for long contexts. **(a, b)** We construct two tasks to perform controlled long-context analysis: (a) bug localization in large code repositories, and (b) anomaly detection in transaction logs. As context length $T$ grows, in-context accuracy drops and thinking tokens show diminishing returns; with the same FLOP budget, qTTT consistently improves performance. **(c)** qTTT shows improvements across domains and model sizes on LongBench-v2 and ZeroScrolls benchmarks.

We design two realistic sandbox tasks to perform controlled experiments and diagnose long-context failure modes. We identify that standard "in-context only" settings fail with growing context length (Figure 1). We formalize this as a limitation of static, finite-precision self-attention, and term it *score dilution*: In presence of "distractor" tokens, logit on a "target" is insufficiently separated from the distractor logits, weakening the target probability mass. We establish that as context length $T$ grows, the target–distractor logit margin must scale as $\Omega(\log T)$ to avoid vanishing target probability. We extend this analysis to show that vanilla compute-scaling strategies, such as "thinking" tokens, cannot retrieve the signal from buried target tokens.

Hence, a natural question arises: *How can we best use inference-time compute to improve long-context retrieval and reasoning?* We revisit test-time training (TTT) (Liu et al., 2021; Hardt & Sun, 2024; Akyürek et al., 2024) as a way to adapt the model to a given long-context input rather than produce more text from an unchanged model. Our key idea, *query-only TTT* (*qTTT*), is a computationally frugal approach: Perform a single prefill to cache keys and values, followed by a few **lightweight gradient updates exclusively on the query projection matrices** in the attention layers, keeping all other parameters fixed and reusing the key-value cache (Figure 2). We show theoretically that this targeted adaptation directly increases the separation between target and distractor logits for the specific context at hand, countering the limitations of vanilla in-context learning.

We perform evaluations on 15+ real-world datasets from popular long-context benchmarks, ZeroScrolls (Shaham et al., 2023) and LongBench-v2 (Bai et al., 2023b), with Qwen3 models spanning 1.7B–8B parameters. We observe consistently large performance gains across model sizes and datasets. Under FLOP-matched inference-time compute budgets, qTTT consistently surpasses standard inference-time thinking strategies (Figure 1c) with more than 20% improvements on code comprehension, multi-document QA, and other multi-hop reasoning tasks. Our results call for reallocating inference-time budget from thousands of "thinking" tokens to a small number of query updates for long-context retrieval and reasoning without altering pre-training, architecture, or data.

**Contributions.**

- We construct sandbox tasks to demonstrate long-context failure modes (§2.1). We formalize *score dilution* in static, finite-precision self-attention and prove a *logarithmic margin requirement*: the target–distractor logit gap must scale as $\Omega(\log T)$ to avoid vanishing target probability (§2.3).

- We show theoretically and empirically that current inference-time compute scaling strategies primarily scale decoding and cannot reliably meet the margin requirement; in particular, they cannot amplify the signal from buried targets beyond an $\varepsilon$-fraction (§2).

- We introduce query-only TTT (qTTT): a compute-frugal TTT procedure that performs one prefill to cache K/V, then applies a few gradient updates *only* to query projections while reusing the KV cache, directly increasing target–distractor separation (§3).

- On 15+ real-world datasets from ZeroScrolls and LongBench-v2, using Qwen3 models (1.7B–8B), query-only TTT consistently improves long-context performance and under FLOP-matched budgets, outperforms intermediate thinking-token baselines (Figure 1c; §4).

Since qTTT takes place at inference-time, it can easily be applied on top of other existing strategies for long-context modeling: architectural changes such as sliding window attention (Dai et al., 2019; Beltagy et al., 2020), adaptive positional encoding (Press et al., 2022; Su et al., 2024), training tweaks for longer windows (Chen et al., 2023; Peng et al., 2024), or retrieval augmented generation (Borgeaud et al., 2022; Izacard et al., 2022).

## 2 VANILLA COMPUTE-SCALING STRATEGIES FAIL FOR LONG CONTEXTS

In this section, we analyze how increasing context length $T$ affects static quadratic-attention LLMs and common inference-time compute–scaling strategies. Using controlled synthetic tasks that mirror realistic long-context retrieval, we observe sharp performance degradation as $T$ grows, while generating intermediate "thinking" tokens yields rapidly diminishing returns. We then provide a theoretical explanation: with static, finite-precision self-attention, the target logit suffers *score dilution* as distractors accumulate, and avoiding this requires a *logarithmic margin requirement*—the worst-case target–distractor logit gap must scale as $\Omega(\log T)$. Decoding-based inference strategies do not reliably meet this requirement; in contrast, small gradient-based adaptations can increase the margin, which motivates our methodology (developed in §3). All proofs are provided in Appendix B.

### 2.1 EMPIRICAL ANALYSIS ON SYNTHETIC LONG-CONTEXT TASKS

First, we empirically analyze the effect of context length on vanilla transformer models and current inference-time compute-scaling strategies. We study two synthetic retrieval tasks that mirror realistic long-context use cases while allowing control over the context length $T$. For each example, the relevant evidence ("needle") is held fixed and only the surrounding "haystack" grows, isolating the effect of length on retrieval. We provide examples from our datasets in Appendix A.

**Bug Localization in a Code Repository.** Starting from a large open-source repository[1], we inject a single-line logical bug and ask the model to identify and fix it. Examples of bugs include missing softmax temperature scaling in the attention mechanism and layernorm misplacement in the Transformer block (see Appendix for details). We vary the context length by the number of lines $L$ exposed to the model. For a given bug instance, we sample a span of $L$ lines around the bug, extending to other files in the directory for large $L$. We create splits of the dataset with $L$ ranging from 5 to 10000. Across length conditions, the bug location and content are held fixed; only the surrounding code (the "haystack") grows to introduce realistic, semantically relevant distractors.

**Error in a Log of Transactions.** We synthesize multi-account banking logs with an initial state and a sequence of operations, each line recording old→new balances and indexed with a TX_ID. Valid logs must satisfy invariants: conservation of total funds, non-negative balances, and arithmetic correctness. We inject exactly one anomaly and consider the following bug types: CALC_ERROR (incorrect arithmetic), NEGATIVE_BAL (over-debit), LOST_UPDATE (stale write overwrites a prior commit) and DUPLICATE_TXN (same payment applied twice). The model must output the bug type and offending TX_ID. Context length is controlled by the number of operations $n$; we sweep from 25 operations to 500 operations which varies the number of tokens from $\mathcal{O}(10^2)$ to $\mathcal{O}(10^4)$.

**Findings.** We evaluate Qwen3 models ranging from 1.7B to 8B parameters on these synthetic tasks. Figure 1 shows the results for the Qwen3-4B model. For both tasks, we see clear consistent trends: (i) As the context lengths increases (number of code lines/transaction logs), the standard in-context performance (i.e., without any additional inference-time compute) decreases sharply. (ii) Further, using inference-time compute via thinking tokens improves performance for shorter contexts, but shows clear diminishing returns as the context length increases, asymptotically converging close to the standard model performance for long contexts.

---

[1]We use OLMo as a reference repository for the dataset: `https://github.com/allenai/OLMo`.

> **Empirical Takeaway:** Across both controlled tasks, holding the needle fixed and increasing the haystack length $T$ yields a sharp, monotonic drop in *in-context* accuracy. Allocating inference-time budget to "thinking" tokens offers only short-horizon gains with clear saturation at large $T$. These trends suggest a structural limitation of static attention in long contexts.

We now formalize this limitation as *score dilution* and derive the resulting *logarithmic margin requirement*, which explains why decoding-based scaling fails to recover retrieval (§2.3).

## 2.2 PRELIMINARIES

Recall, for a sequence of $T$ tokens with hidden representations $\{h_i\}_{i=1}^T \in \mathbb{R}^d$, each Transformer layer $\ell$ computes query, key, and value projections:

$$q_i^{(\ell)} = W_Q^{(\ell)} h_i, \quad k_j^{(\ell)} = W_K^{(\ell)} h_j, \quad v_j^{(\ell)} = W_V^{(\ell)} h_j, \tag{2.1}$$

where $W_Q^{(\ell)}, W_K^{(\ell)} \in \mathbb{R}^{d_k \times d}$ and $W_V^{(\ell)} \in \mathbb{R}^{d_v \times d}$ are learned projection matrices. Further, the scaled dot product between query $q_i$ and key $k_j$ gives the attention logits $z_{i,j}$ that are normalized via softmax to obtain attention weights $\alpha_{i,j}$. Finally, the output $o_i$ is a weighted sum of value vectors:

$$z_{i,j} := \frac{q_i^\top k_j}{\sqrt{d_k}}, \qquad \alpha_{i,j} := \frac{\exp(z_{i,j})}{\sum_{\ell=1}^T \exp(z_{i,\ell})}, \qquad o_i = \sum_{j=1}^T \alpha_{i,j} v_j. \tag{2.2}$$

In the autoregressive setting, causal masking enforces $j \leq i$, so that each position $i$ can only aggregate information from its past. Multi-head attention extends this computation across several subspaces, allowing the model to capture diverse forms of dependency.

**In-Context Learning.** This attention-based retrieval is the foundation of *in-context learning* (ICL; (Dong et al., 2023)). By inserting task demonstrations, instructions, or relevant passages directly into the input, LLMs can adapt their outputs without parameter updates. For applications such as analyzing codebases, synthesizing long documents, or sustaining multi-turn dialogues, the model must effectively identify and use information scattered across contexts of length $10^4$–$10^6$ tokens.

**Thinking Tokens.** Given a prefix $x_{1:i}$ and a target at position $i+1$, *thinking-token* methods (Wei et al., 2022a; Kojima et al., 2022; Wang et al., 2023a) append $M \geq 0$ auxiliary tokens at indices $t \in \{i+1, \ldots, i+M\}$ before producing the final answer at $a = i+M+1$. Each token $t$ is generated with static parameters and the same attention kernel as in Equation (2.2), yielding logits $z_{t,j}$, weights $\alpha_{t,j}$, and outputs $o_t$ over the augmented sequence of length $T' = T+M$.

**Definition 2.1** (Retrieval). When predicting token $x_{i+1}$, the relevant information may lie in a specific key–value pair $(k_{j^*}, v_{j^*})$ (the '*needle*') at some earlier position $j^* < i$. For a threshold $\tau \in (0, 1)$, we say that retrieval at position $i$ succeeds if $\alpha_{i,j^*} \geq \tau$. Equivalently, in margin form define $\gamma_i := z_{i,j^*} - \log\sum_{j \neq j^*} e^{z_{i,j}}$, then retrieval succeeds iff

$$\gamma_i \geq \log\left(\frac{\tau}{1-\tau}\right).$$

All other positions $j \neq j^\star$ are *distractors*, contributing competing logits $\{z_{i,j}\}_{j \neq j^\star}$.

## 2.3 THEORETICAL LIMITATIONS OF STATIC ATTENTION AND THINKING TOKENS

Informed by the empirical findings in §2.1, we now analyze a single attention layer as in Equation (2.2) on the retrieval task (Definition 2.1). We formalize the fundamental challenge of score dilution, which arises when "near-tie" distractors inflate the softmax denominator, causing even a unique maximal logit to receive vanishingly small attention mass.

**Lemma 2.2** (Score dilution). *If at least $m$ distractor keys satisfy $z_{i,j} \geq z_{i,j^*} - \Delta$ for some $\Delta \geq 0$, then*

$$\alpha_{i,j^\star} \leq \frac{1}{1 + me^{-\Delta}}.$$

*In particular, if $m \geq cT$ for some $c > 0$ and $\Delta = O(1)$, then $\alpha_{i,j^\star} \to 0$ as $T \to \infty$.*

This lemma formalizes a simple intuition: When a constant fraction of tokens are within $O(1)$ logit of the needle, the attention budget cannot concentrate and the needle's mass vanishes with $T$.

This dilution effect imposes a strict requirement on how much the target logit must stand out from all other distractors. The following corollary quantifies this necessary separation, showing that the required margin between needle and distractor must grow with the context length.

**Lemma 2.3** (Logarithmic margin requirement). *Fix $\varepsilon \in (0,1)$. If*

$$\min_{j \neq j^\star} \left( z_{i,j^\star} - z_{i,j} \right) \ \geq \ \log\Big( \frac{(T-1)(1-\varepsilon)}{\varepsilon} \Big),$$

*then $\alpha_{i,j^\star} \geq 1 - \varepsilon$. In particular, guaranteeing a fixed target mass against worst-case distractors requires a gap that scales as $\Omega(\log T)$.*

Achieving a margin that scales logarithmically is difficult for a model with static attention. Next, we evaluate the strategy of generating thinking tokens in satisfying the logarithmic margin requirement.

**Proposition 2.4** (Needle-signal bound for generated tokens). *For any thinking token $t \in \{i+1, \ldots, i+M\}$ and any $u \in \mathbb{R}^{d_v}$,*

$$\langle u, o_t \rangle \ \leq \ \alpha_{t,j^\star} \langle u, v_{j^\star} \rangle \ + \ \left( 1 - \alpha_{t,j^\star} \right) \max_{j \neq j^\star} \langle u, v_j \rangle.$$

**Corollary 2.5** (Specialization under small margin). *If the margin at token $t$ satisfies $\gamma_t \leq \log\big(\varepsilon/(1-\varepsilon)\big)$ (equivalently, $\alpha_{t,j^\star} \leq \varepsilon$ by Definition 2.1), then*

$$\langle u, o_t \rangle \ \leq \ \varepsilon \langle u, v_{j^\star} \rangle \ + \ (1 - \varepsilon) \max_{j \neq j^\star} \langle u, v_j \rangle.$$

*Moreover, by Lemma 2.2, if at least $m$ distractors satisfy $z_{t,j} \geq z_{t,j^\star} - \Delta$, then $\alpha_{t,j^\star} \leq 1/(1 + me^{-\Delta})$, yielding the same bound with $\varepsilon = 1/(1 + me^{-\Delta})$.*

Proposition 2.4 shows the fraction of needle signal any generated token can carry is *at most* its own attention mass on the needle. Under dilution (small margin), this mass is provably tiny (Corollary 2.5), so attending to thinking tokens cannot materially increase the final answer's effective margin unless some intermediate token first assigns non-trivial attention to the needle.

> **Takeaways:** **(i)** With fixed weights, worst-case retrieval requires a logit margin that grows like $\Omega(\log T)$; failing to achieve this leads to score dilution and vanishing $\alpha_{i,j^\star}$. **(ii)** Autoregressively generating additional tokens with the same static attention does not repair missing access to the evidence. **(iii)** Any successful inference-time strategy must change the similarity $q_i^\top k_j$ (e.g., by updating queries) rather than sampling more tokens with unchanged parameters.

## 3 EFFICIENT TEST-TIME ADAPTATION VIA QUERY-ONLY UPDATES

Having established that existing inference-time scaling strategies on vanilla transformer models fail for long contexts, we now investigate an alternate strategy of allocating inference-time compute via test-time training (TTT). First, we establish why a standard TTT approach, involving several forward and backward passes over the model, is computationally infeasible for long contexts. We introduce query-only TTT (qTTT) that captures the benefits of TTT while minimizing the computational overhead by re-using the KV cache and only changing the query projections. We present theoretical (§3.2) and empirical (§4) evidence for the efficacy of qTTT over vanilla ICL and thinking tokens.

**Naïve Test-Time Training is Infeasible for Long Contexts.** A natural first-step is full-parameter TTT: update FFN and all attention projections ($W_Q, W_K, W_V$) on the long input $x_{1:T}$. We find that this is impractical for long-context regimes: every update alters keys/values across the sequence, invalidating the KV cache and forcing fresh forward–backward passes over the *entire* context at each step, with prohibitive compute and activation memory.

Compute-wise, our FLOP calculations (Appendix C) shows that even *one* such full-parameter TTT step over a $T$-token context is equivalent to generating about $1.2 \times T$ decoding tokens. That is, for

---

**Algorithm 1** Query-Only Test-Time Training for Long Context

---

1: **Input:** model $f_\theta$, long context $x_{1:T}$, number of steps $N_{\text{TTT}}$, span length $k$, step size $\eta$
2: $\{K^{(\ell)}, V^{(\ell)}\}_{\ell=1}^L \leftarrow \text{FORWARDPASSANDCACHE}(f_\theta, x_{1:T})$     $\triangleright$ Single $O(T^2)$ operation
3: **for** $n = 1$ to $N_{\text{TTT}}$ **do**
4:     Sample a random span $x_s = x_{t:t+k}$ from $x_{1:T}$
5:     Compute $\mathcal{L}_{\text{TTT}}(\theta; x_s)$ using the frozen $\{K^{(\ell)}, V^{(\ell)}\}$
6:     Update only the query parameters: $\{W_Q^{(\ell)}\} \leftarrow \{W_Q^{(\ell)}\} - \eta \, \nabla_{\{W_Q^{(\ell)}\}} \mathcal{L}_{\text{TTT}}$
7: **end for**
8: **return** adapted model $f_{\theta'}$ to generate the final answer

---

a context of about $T \approx 10^5$ tokens, this makes a single training step FLOP equivalent to generating $\sim 120\text{K}$ decoding tokens—rendering full-parameter TTT untenable.

These constraints motivate a cache-preserving alternative. Our approach, query-only TTT (qTTT), performs a single prefill to cache $\{K, V\}$ and then adapts *only* the query projections on short spans, keeping the attention evidence pathway fixed while reshaping access to it. This retains the benefits of TTT without repeated full-context passes; we describe and formalize this procedure next.

### 3.1 Query-Only TTT for Long Context

The core idea of query-only TTT is to avoid repeated, costly forward and backward passes over the long context. Instead, we perform a single expensive prefill to cache the context's key and value representations and then execute a series of much cheaper, targeted gradient updates. The procedure, also outlined in Algorithm 1 and Figure 2, is as follows:

1. **Single-Pass KV Cache Generation.** Given a long context $x_{1:T}$, we perform exactly one full forward pass with the pre-trained model $f_\theta$. During this pass, for each layer $\ell$ in the model, we compute and store the Key and Value projection tensors, $K^{(\ell)} \in \mathbb{R}^{T \times d_k}$ and $V^{(\ell)} \in \mathbb{R}^{T \times d_v}$. These cached tensors represent the complete contextual information and remain frozen for the duration of the adaptation process.

2. **Span-Sampled, Query-Only Objective.** With the KV cache held constant, we perform $N_{\text{TTT}}$

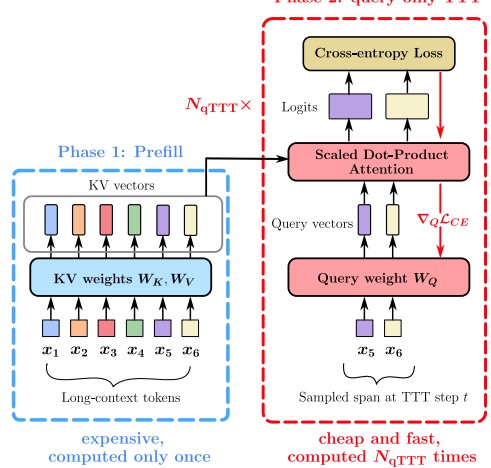

Figure 2: Overview of query-only TTT.

steps of gradient descent. In each step, we update only the query projection matrices $\{W_Q^{(\ell)}\}_{\ell=1}^L$. The objective is the standard next-token prediction loss, computed over a small, randomly sampled contiguous span of tokens $x_s = x_{t:t+k}$, where the span length $k \ll T$:

$$\mathcal{L}_{\text{TTT}}(\theta; x_s) = -\sum_{i=t}^{t+k-1} \log p_\theta(x_{i+1} \mid x_{1:i}; \{K^{(\ell)}, V^{(\ell)}\}_{\ell=1}^L) \tag{3.1}$$

Crucially, the gradients $\nabla_\theta \mathcal{L}_{\text{TTT}}$ are computed and applied only with respect to the parameters $\{W_Q^{(\ell)}\}$, leaving all other model weights, including the now-static KV cache, unchanged.

### 3.2 Why Query-Only Test-Time Training is Effective

Section 2 showed that long-context failures arise from score dilution and the resulting need for a growing target–distractor *margin*. Query-only TTT targets this bottleneck directly: only adapt the query projections while holding keys/values fixed (from a single prefill). This leaves the evidence (K,V) unchanged and instead reshapes *query* to it by modifying the similarity $q_i^\top k_j$ for a given input (Proposition 3.1; Figure 3).

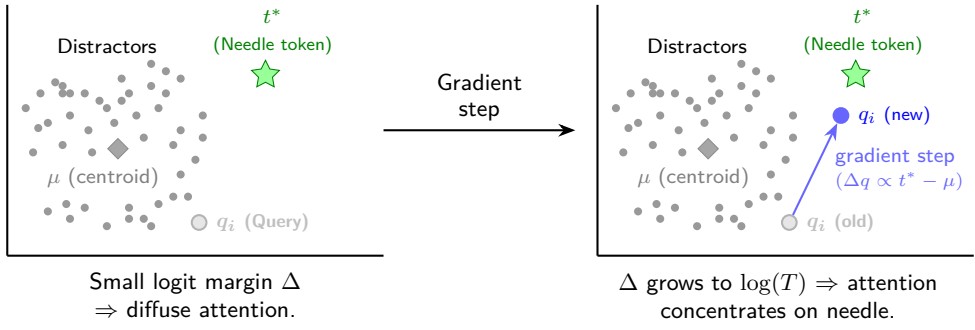

Figure 3: A visual representation of Proposition 3.1 showing how qTTT improves the logit margin. The gradient updates via qTTT directly move the query projection weights towards the target needles and counteracts score dilution.

**Proposition 3.1** (Query update). *For loss $\ell_i = -\log \alpha_{i,j^\star}$ with fixed $K$, the gradient w.r.t. $q_i$ is*

$$\nabla_{q_i}\ell_i = \frac{1}{\sqrt{d_k}}\Big(\underbrace{\sum_{\ell=1}^{T} \alpha_{i,\ell}k_\ell}_{\mu_i} - k_{j^\star}\Big).$$

*A descent step $q_i \leftarrow q_i - \eta\nabla_{q_i}\ell_i$ moves $q_i$ toward $k_{j^\star}$ and away from the attention-weighted mean $\mu_i$, explicitly counteracting dilution. (The statement holds per head and aggregates across heads.)*

**Lemma 3.2** (Margin improvement). *Let $M_i(q_i) := -\ell_i(q_i)$ denote the logit margin. For sufficiently small $\eta > 0$,*

$$M_i\big(q_i - \eta\nabla_{q_i}\ell_i\big) = M_i(q_i) + \eta\|\nabla_{q_i}\ell_i\|_2^2 + O(\eta^2).$$

*Hence the margin strictly increases whenever $\nabla_{q_i}\ell_i \neq 0$, with the gain proportional to $\|k_{j^\star} - \mu_i\|_2^2$. Improvements are therefore largest precisely when attention is most diffuse, i.e., in the long-context regimes where score dilution is severe.*

> **Takeaway:** Query-only TTT reallocates inference-time compute into *margin-raising* updates: with fixed $\{K, V\}$ from a single prefill, each step moves $q_i$ toward $k_{j^\star}$ and *provably* increases the target–distractor logit margin. It thus directly mitigates score dilution, most when attention is most diffuse, without re-encoding the context or growing the KV cache.

### 3.3 FLOP EQUIVALENCE: THINKING TOKENS VS. QUERY-ONLY TTT

We compare two ways to spend inference-time compute after a single prefill: (i) generate $T_{\text{think}}$ *thinking* tokens with frozen weights, or (ii) run $N_{\text{qTTT}}$ *query-only* updates on spans of length $k \ll T$ while reusing the KV cache. For long $T$, FLOP equivalence (Appendix C) yields the rule of thumb

$$T_{\text{think}} \approx 2\,N_{\text{qTTT}}\,k \qquad \text{(long } T, \text{ span } k \ll T\text{).} \tag{3.2}$$

Consider a dense model of about 8B parameters on a long context $T = 10^5$ and an inference-time budget budget to decode 8K thinking tokens after the prefill. From Equation (3.2), the FLOPs equate to about $N_{\text{qTTT}} = 16$ query-only TTT steps on spans of $k = 128$, and $N_{\text{qTTT}} = 8$ for $k = 512$. In both cases, thinking tokens grow the KV cache by thousands of positions without changing attention, whereas query-only TTT keeps the cache length fixed at $T$ and uses the matched FLOPs to *reshape queries* against the existing keys/values, directly targeting the margin bottleneck from §2.

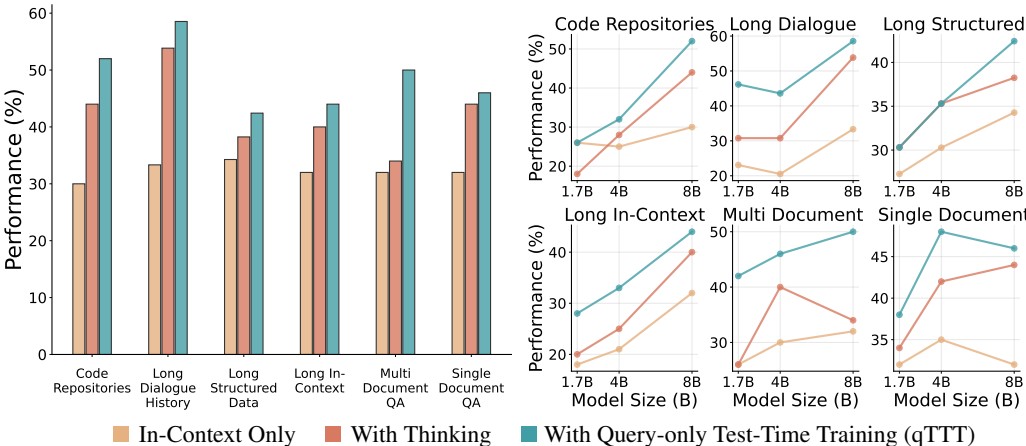

(a) Comparison on LongBench-v2 subsets for Qwen3-8B. Using qTTT consistently outperforms standard in-context and FLOP-matched thinking settings.

(b) Variation of performance across model size on LongBench-v2 subsets. qTTT improves performance consistently across model sizes.

Figure 4: LongBench-v2 (Bai et al., 2023b) provides a testbed to evaluate long-context abilities across a diverse set of context types. Here, we report evaluations across all six subsets of the benchmark for Qwen3-{1.7/4/8B} models. qTTT shows consistent improvements over both standard in-context learning and FLOP-matched thinking tokens across the different context types.

## 4 EXPERIMENTAL RESULTS

In this section, we discuss experimental results across a suit of long-context tasks. Firstly, we call-back the synthetic long-context setup from §2.1. Figure 1 shows that spending inference-time compute via query-only TTT results in significant performance improvements on top of just in-context decoding. We observe that the improvements are consistent across context lengths unlike thinking tokens that show rapid diminishing returns. In the rest of this section, we discuss our findings on long-context benchmarks that involve nuanced $n$-hop retrieval, reasoning, and comprehension.

Further, we empirically verify that these improvements with qTTT are indeed a result of margin improvement and reducing score dilution. Appendix E (Table 2) shows an analysis of attention mass on the target tokens with and without qTTT. Particularly, we aggregate the attention scores for the target tokens (well defined for these synthetic tasks) across model layers to study the influence of qTTT against vanilla attention. We observe that as number of input tokens increases (hence the number of distractors), the performance as well as attention mass for vanilla attention goes down drastically. However, qTTT helps preserve attention mass significantly across context lengths.

**Setup and Evaluation Protocol.** We evaluate query-only TTT (qTTT) on long-context tasks against two baselines: (i) *In-context*—standard decoding with no intermediate tokens; and (ii) *Thinking*—a chain-of-thought variant whose extra tokens are *compute-matched* to qTTT via the FLOP equivalence in §3.3. Our experiments are performed over Qwen3 models across 1.7B, 4B, and 8B parameters, and cover all subsets of **LongBench-v2** (Bai et al., 2023b) (six categories) and **ZeroS-rolls** (Shaham et al., 2023) (eight datasets). Unless stated otherwise, we use $T_{\text{think}}=8192$, $k=128$, $N_{\text{qTTT}}=32$, and a common budget of $512$ tokens to generate the final answer[2].

**LongBench-v2.** LongBench-v2 (Bai et al., 2023b) evaluates long-context reasoning across diverse context types. The benchmark probes whether models can locate and use dispersed evidence to answer multiple-choice questions across a variety of context types: given multi-file project trees in the *Code Repositories* setting, to resolve arguments of a particular function; and given the context as a set of related documents in the *Multi-Document QA* setting, synthesize spans across sources to answer a question. This allows us to assess the applicability of qTTT across forms of input contexts.

---

[2]We use the /think and /no_think tokens in the Qwen3 model to control for this. We elaborate on further details including decoding parameters and prompt templates in Appendix D.

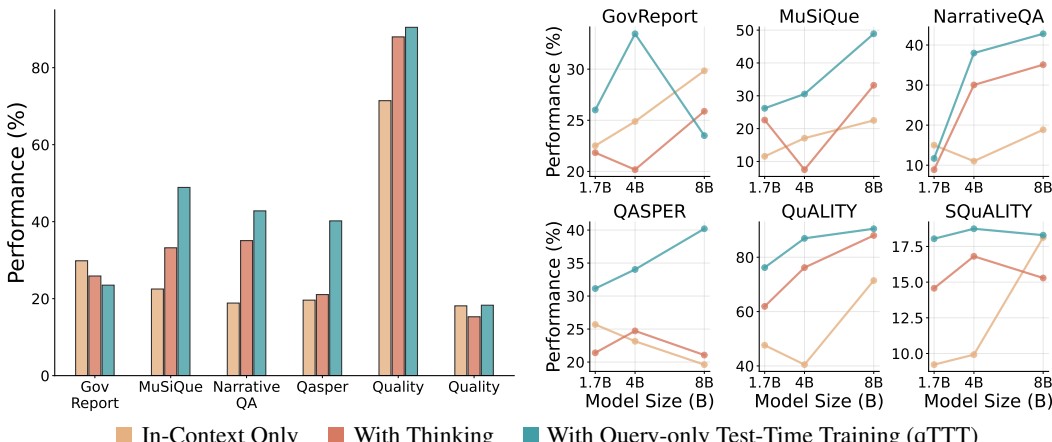

(a) Comparison on ZeroScrolls subsets for Qwen3-8B. Using qTTT consistently outperforms standard in-context and FLOP-matched thinking settings.

(b) Variation of performance across model size on ZeroScrolls subsets. qTTT improves performance consistently across sizes, often greater for larger models.

Figure 5: ZeroScrolls (Shaham et al., 2023) evaluates a diverse set of tasks and model abilities over long context inputs. We report evaluations across six subsets for Qwen3-{1.7/4/8B} models. qTTT shows consistent improvements over both standard in-context learning and FLOP-matched thinking tokens, especially for retrieval-based multi-hop reasoning and long form comprehension tasks.

Figure 4 shows that, under compute-matched budgets, qTTT delivers consistent and often substantial gains across model sizes. On *Long Dialogue History* and *Multi-Document QA*, where evidence is most diffuse, qTTT outperforms standard in-context and thinking by wide margins (e.g., for Qwen3-4B: 30.8 → **43.6** on *Long Dialogue History*; 40.0 → **46.0** on *Multi-Document QA*). In *Code Repositories*, qTTT scales especially well with model size (for Qwen3-8B: 30.0 → 44.0 → **52.0**). Overall, the LongBench-v2 results indicate that qTTT fares well across markedly different context types.

**ZeroScrolls.** ZeroScrolls (Shaham et al., 2023) evaluates long-context reasoning across diverse tasks. We group the datasets into three categories: (i) *Multi-hop reasoning* (MuSiQue, QASPER, NarrativeQA), which require locating and composing evidence spread across long documents; (ii) *Long-form summarization* (GovReport, QMSum, SQuALITY), which emphasize distilling lengthy inputs; and (iii) *Long-passage comprehension* (QAuLITY), which measures multiple-choice accuracy over extended contexts. In contrast to LongBench-v2, this suite of tests evaluates the ability to utilize some long context to solve a variety of different tasks.

Figure 9 shows that qTTT consistently outperforms vanilla thinking on multi-hop QA and comprehension tasks, with gains that strengthen with model size. On summarization-style datasets, improvements are smaller and comparable to thinking, suggesting that when generation quality, not retrieval, is the primary bottleneck, reweighting attention yields limited returns. Overall, we see significant performance gains across datasets and model scales.

The full set of results on LongBench-v2 and ZeroScrolls are elaborated in Appendix F. Moreover, we include additional test-time compute baselines such as best-of-N and beam search in Appendix G. We also perform a comprehensive latency and wall-clock time comparison of qTTT with other approaches in Appendix H.

> **Takeaways:** **(i)** We see consistent gains in performance across benchmarks and model sizes, qTTT yields the best average performance under matched FLOPs (Figure 8, Figure 9). **(ii)** Retrieval-driven tasks benefit the most, validating the score dilution diagnosis and the margin increase with qTTT (§2, §3.2). **(iii)** Thinking tokens are not a reliable substitute: they sometimes help but can also trail *In-context*, especially in very long contexts. **(iv)** qTTT is a more effective use of inference-time compute; without changing architecture, data, or pre-training.

## 5 PRIOR WORK

**Long-context LLMs.** Context windows have expanded rapidly, with models reaching million-token scale (Reid et al., 2024), usually extending limits via RoPE scaling (Chen et al., 2023; Bai et al., 2023a). Parallel efforts reduce quadratic attention with sparse/structured patterns (Beltagy et al., 2020; Zaheer et al., 2020). Evaluation has coalesced around long-context suites such as LongBench/LongBench-v2 (Bai et al., 2023b), ZeroScrolls (Shaham et al., 2023), RULER, and domain-specific code benchmarks like SWE-bench variants (Jimenez et al., 2024). However, these LLMs still exhibit strong position sensitivity, yielding the "lost in the middle" effect (Liu et al., 2024). Needle-in-a-haystack–style tests show that a single relevant span can be overwhelmed by many distractors, and this persists across languages and document structures (Kamradt, 2024). Our work targets this retrieval failure by addressing how attention mass is allocated over very long inputs.

**Inference-time compute scaling.** A common approach is to spend more compute at inference via chain-of-thought (Wei et al., 2022c), self-consistency (Wang et al., 2023b), best-of-$n$ selection (Nakano et al., 2021), or other strategies (Zelikman et al., 2024; Zweiger et al., 2025; Kang et al., 2025). While often helpful, these methods scale decoding and can be compute-heavy with diminishing returns (Snell et al., 2024; Liu et al., 2025). Another way to spend inference-compute is via test-time training (Sun et al., 2020; Liu et al., 2021; Hardt & Sun, 2024; Akyürek et al., 2024). While typically done handle distribution shift, recent work has started focusing on long-context LLM use cases (Sun et al., 2024; Zuo et al., 2025). To our knowledge, our work is first re-purpose TTT to the micro-distribution of individual inputs via an efficient, query-only variant tailored to long-context.

## 6 DISCUSSION

We identify score dilution in static quadratic attention as a core cause of long-context failures. We design synthetic tasks to study long-context behavior controllably and show that accuracy falls sharply with context length $T$ and "thinking" tokens show diminishing returns (§2). We proposed query-only TTT (qTTT) to reallocate inference-time budget via few query-only updates that provably increase the target–distractor margin (§3). Under matched FLOPs, qTTT consistently outperforms *in-context* and *thinking* on LongBench-v2 and ZeroSCROLLS, with the largest gains on retrieval and multi-hop reasoning (§4). In short, adapting queries is a more effective use of inference-time compute than generating more tokens for long context tasks.

**Future directions.** (1) We evaluate a single point on the $(k, N_{\text{TTT}})$ trade-off; exploring budget schedules across span size and steps is immediate. (2) Our compute-matched baseline focuses on "thinking" tokens; extending to self-consistency and best-of-$n$ within the same framework is future work. (3) Gains are task-dependent; developing simple predictors for when to prefer qTTT (vs. decoding-based scaling) is a practical next step.

## 7 ACKNOWLEDGMENTS

This work was done when RB, RT, SSD, and DK were summer interns at Meta. RB would like to thank other interns in the legacy GenAI team for the exchange of ideas and brainstorming that shaped this project. Namely: Irene Zhang, Winnie Yang, Julian Coda-Forno, Sriyash Poddar, Arushi Rai, and others in the Research Club. We thank Sharan Narang, Prateek Yadav, and Mike Lewis for their guidance. RB would like to thank Yonatan Belinkov, Nihal Nayak, Lyndon Lam, Sunny Qin, Bingbin Liu, and other members of the ML Foundations group and the Kempner Institute at Harvard for their feedback on the manuscript.

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

**Bug Description:** The attention mechanism fails to properly normalize attention scores, leading to numerical instability and gradient explosion during training. The attention weights grow unbounded, causing immediate training divergence.

**Code context:**

```
olmo/model.py
L335: def _scaled_dot_product_attention(
L336:     self,
L337:     q: torch.Tensor,
L338:     k: torch.Tensor,
L339:     v: torch.Tensor,
L340:     attn_mask: Optional[torch.Tensor] = None,
L341:     dropout_p: float = 0.0,
L342:     is_causal: bool = False,
L343: ) -> torch.Tensor:
L344:
L345:     attn_weights = torch.matmul(q, k.transpose(-2, -1))
L346:
L347:     if is_causal:
L348:         assert attn_mask is None
L349:         query_len, key_len = q.shape[-2], k.shape[-2]
```

**Target:** Given the above code context, please identify the exact location of the bug.

– – – – – – – – – – – – – – – – – – – – – – – – – – – – – – – – – –

**Model output:** `olmo/model.py:L345`

Figure 6: An example of the code bug localization synthetic task.

## A   SYNTHETIC TASKS

We illustrate two representative synthetic tasks used in our study. Figure 6 shows the *code bug localization* task: the model receives a brief natural-language bug description together with a minimal, line-numbered code context and must return the exact file-and-line of the offending statement. In the example, the model correctly identifies the line where attention scores are computed without proper normalization (`olmo/model.py:L345`).

Figure 7 shows the *transaction-log consistency* task: given an initial account state, a set of invariants (e.g., conservation of total funds, no negative balances), and a short sequence of transfers, the model must select a single bug type and pinpoint the first offending transaction. In the example, the model outputs `NEGATIVE_BAL` at `TX004`, where the balance of account A becomes negative, violating the stated rules.

Together, these examples illustrate the input/output format of our synthetic tasks, the kind of structured context provided to the model, and the expected concise targets (a specific line for code or a {`bug_type`, `TX_id`} pair for logs). We use similarly formatted instances throughout our evaluation.

## B   PROOFS FOR SECTION 2

**Notation.**   For a fixed query $q_i$, logits are $z_{i,j} = \frac{q_i^\top k_j}{\sqrt{d_k}}$, attention weights $\alpha_{i,j} = \frac{e^{z_{i,j}}}{\sum_\ell e^{z_{i,\ell}}}$, and $o_i = \sum_j \alpha_{i,j} v_j$. We write $\mu_i = \sum_\ell \alpha_{i,\ell} k_\ell$.

**Task Description:** Analyze this banking transaction log for bugs.

**Initial state:** {`"account_A"`: 4000, `"account_B"`: 4200, `"total"`: 8200}

**Rules:**
1. Total money must remain constant (conservation)
2. No account can go negative
3. All calculations must be mathematically correct

**Transaction logs:**
```
[TX001]: Transfer $107: A=4000 → 3893, B=4200 → 4307
[TX002]: Transfer $204: A=3893 → 3689, B=4307 → 4511
[TX003]: Transfer $780: A=3689 → 2909, B=4511 → 5291
[TX004]: Transfer $2925:A=2909 → -16,  B=5291 → 8216
[TX005]: Transfer $699: B=8216 → 7517, A=-16  →  683
```

**Possible bug types** (choose exactly one):
 – CALC_ERROR: Mathematical calculation is incorrect
 – NEGATIVE_BAL: Account balance becomes negative
 – LOST_UPDATE: Concurrent update causes lost transaction
 – DUPLICATE_TXN: Same transaction processed multiple times

**Target**: Please identify the bug type and location.
– – – – – – – – – – – – – – – – – – – – – – – – – – – – – – – – – – – –
**Model output:** {`"bug_type"`: NEGATIVE_BAL, `"bug_location"`: TX004}

Figure 7: An example of the log transactions synthetic task.

*Proof of Lemma 2.3 (Score dilution).* Let $S = \{j \neq j^\star : z_{i,j} \geq z_{i,j^\star} - \Delta\}$ with $|S| = m$. Then

$$\sum_{\ell=1}^{T} e^{z_{i,\ell}} \geq e^{z_{i,j^\star}} + \sum_{j \in S} e^{z_{i,j}} \geq e^{z_{i,j^\star}}\big(1 + me^{-\Delta}\big),$$

hence $\alpha_{i,j^\star} = \frac{e^{z_{i,j^\star}}}{\sum_\ell e^{z_{i,\ell}}} \leq \frac{1}{1+me^{-\Delta}}$. If $m \geq cT$ with $c > 0$ and $\Delta = O(1)$, then $\alpha_{i,j^\star} \to 0$ as $T \to \infty$. □

*Proof of Lemma 2.3 (Logarithmic margin requirement).* Let $\gamma = \min_{j \neq j^\star}(z_{i,j^\star} - z_{i,j})$. Then $\sum_{j \neq j^\star} e^{z_{i,j}} \leq (T-1)e^{z_{i,j^\star} - \gamma}$, so

$$\alpha_{i,j^\star} = \frac{1}{1 + \sum_{j \neq j^\star} e^{z_{i,j} - z_{i,j^\star}}} \geq \frac{1}{1 + (T-1)e^{-\gamma}}.$$

Rearranging $\frac{1}{1+(T-1)e^{-\gamma}} \geq 1 - \varepsilon$ yields $\gamma \geq \log\big(\frac{(T-1)(1-\varepsilon)}{\varepsilon}\big)$. □

*Proof of Proposition 2.4 (Needle-signal bound).* For any thinking token $t$,

$$o_t = \sum_{j<t} \alpha_{t,j} v_j = \alpha_{t,j^\star} v_{j^\star} + (1 - \alpha_{t,j^\star}) \sum_{j \neq j^\star} \tilde{\alpha}_{t,j} v_j, \quad \tilde{\alpha}_{t,j} = \frac{\alpha_{t,j}}{1 - \alpha_{t,j^\star}}.$$

For any $u \in \mathbb{R}^{d_v}$, take inner products and upper bound the convex combination by its maximum term:

$$\langle u, o_t \rangle \leq \alpha_{t,j^\star} \langle u, v_{j^\star} \rangle + (1 - \alpha_{t,j^\star}) \max_{j \neq j^\star} \langle u, v_j \rangle.$$

□

*Proof of Corollary 2.5 (Specialization under small margin).* By Definition 2.1, $\gamma_t \leq \log\big(\varepsilon/(1-\varepsilon)\big)$ iff $\alpha_{t,j^\star} \leq \varepsilon$. Substitute $\alpha_{t,j^\star} \leq \varepsilon$ in Proposition 2.4 to obtain

$$\langle u, o_t \rangle \leq \varepsilon \langle u, v_{j^\star} \rangle + (1 - \varepsilon) \max_{j \neq j^\star} \langle u, v_j \rangle.$$

Moreover, Claim 2.3 implies $\alpha_{t,j^\star} \le 1/(1 + me^{-\Delta})$ when at least $m$ distractors satisfy $z_{t,j} \ge z_{t,j^\star} - \Delta$, yielding the bound with $\varepsilon = 1/(1 + me^{-\Delta})$. $\qquad\square$

*Proof of Claim 3.1 (Directional query update).* With $z_{i,\ell} = \frac{q_i^\top k_\ell}{\sqrt{d_k}}$,

$$\ell_i(q_i) = -\log \alpha_{i,j^\star} = -z_{i,j^\star} + \log \sum_{\ell=1}^{T} e^{z_{i,\ell}}.$$

Differentiating w.r.t. $q_i$ and using $\frac{\partial z_{i,\ell}}{\partial q_i} = \frac{k_\ell}{\sqrt{d_k}}$,

$$\nabla_{q_i} \ell_i = -\frac{k_{j^\star}}{\sqrt{d_k}} + \frac{1}{\sum_{\ell'} e^{z_{i,\ell'}}} \sum_{\ell=1}^{T} e^{z_{i,\ell}} \frac{k_\ell}{\sqrt{d_k}} = \frac{1}{\sqrt{d_k}} \Big( \sum_{\ell=1}^{T} \alpha_{i,\ell} k_\ell - k_{j^\star} \Big) = \frac{1}{\sqrt{d_k}} (\mu_i - k_{j^\star}).$$

Thus a descent step moves $q_i$ toward $k_{j^\star}$ and away from $\mu_i$. $\qquad\square$

*Proof of Lemma 3.2 (Monotone margin improvement).* Define $M_i(q_i) = -\ell_i(q_i)$. Then $\nabla M_i(q_i) = -\nabla \ell_i(q_i)$. For a step $q_i^+ = q_i - \eta \nabla \ell_i(q_i)$, a first-order expansion gives

$$M_i(q_i^+) = M_i(q_i) + \eta \|\nabla \ell_i(q_i)\|_2^2 + O(\eta^2).$$

Using Claim 3.1, $\|\nabla_{q_i} \ell_i\|_2^2 = \frac{1}{d_k} \|k_{j^\star} - \mu_i\|_2^2$, which is strictly positive unless $k_{j^\star} = \mu_i$. If $\nabla \ell_i$ is $L$-Lipschitz, choosing $\eta \in (0, 1/L]$ ensures $M_i(q_i^+) \ge M_i(q_i) + \frac{\eta}{2} \|\nabla \ell_i(q_i)\|_2^2$. $\qquad\square$

**Remarks on multi-head attention.** All statements apply per head. Let superscript $h$ index heads and define per-head logits/weights $\{z_{i,j}^{(h)}, \alpha_{i,j}^{(h)}\}$. Claims on dilution and margin hold headwise; aggregation across heads is via concatenation and an output projection, which preserves the directional and margin-improvement arguments by linearity.

## C  FLOP DERIVATIONS FOR §3.3

We outline FLOP models for two inference-time modes and derive the equivalence summarized in Eq. equation 3.2. Consider a dense Transformer with $L$ layers, hidden size $d$, MLP ratio $r$ (so $d_{\text{ff}} = rd$), and long context length $T$. Let $T_{\text{think}}$ be the number of autoregressively generated "thinking" tokens, $N_{\text{qTTT}}$ the number of query-only updates, and $k$ the span size per update.

**Cost coefficients.** Ignoring lower-order terms (layer norms, biases), we collect the dominant costs as

$C_{\text{quad}} = 2Ld$ (quadratic attention term), $\qquad C_{\text{tok}} = (4+2r)Ld^2$ (per-token projections/MLP).

A parallel forward over $T$ tokens (the prefill) costs

$$F_{\text{prefill}}(T) = C_{\text{quad}} T^2 + C_{\text{tok}} T.$$

**Case A (autoregressive "thinking").** After one prefill, generating $T_{\text{think}}$ tokens with a KV cache costs

$$F_{\text{gen}}(T_{\text{think}}; T) = C_{\text{quad}} \Big( T_{\text{think}} T + \frac{T_{\text{think}}(T_{\text{think}} - 1)}{2} \Big) + C_{\text{tok}} T_{\text{think}},$$

so the total is $F_A = F_{\text{prefill}}(T) + F_{\text{gen}}(T_{\text{think}}; T)$.

**Case C (query-only TTT: query-only with cached K/V).** With one prefill, each query-only pass recomputes queries for $k$ positions that attend to cached $\{K, V\}$ and backpropagates only into $\{W_Q\}$. The per-pass cost is

$$G_{\text{partial}}(k; T) \approx 2\Big( C_{\text{quad}} kT + (2+2r)L k d^2 \Big),$$

and the total is $F_C = F_{\text{prefill}}(T) + N_{\text{qTTT}} G_{\text{partial}}(k; T)$. (If the span also attends within itself, add $+C_{\text{quad}} k^2$ and $+2Lkd^2$ inside $G_{\text{partial}}$, which are dominated by $kT$ when $k \ll T$.)

**Equivalence (A vs. C).** Cancelling the shared prefill and equating $F_{\text{gen}}(T_{\text{think}}; T) = N_{\text{qTTT}} G_{\text{partial}}(k; T)$ yields

$$C_{\text{quad}}\left(T_{\text{think}} T + \frac{T_{\text{think}}(T_{\text{think}}-1)}{2}\right) + C_{\text{tok}} T_{\text{think}} = 2N_{\text{qTTT}} k\left(C_{\text{quad}} T + (2+2r)Ld^2\right).$$

For long contexts with $T \gg d$ and spans $k \ll T$ (hence $T_{\text{think}} \ll T$ in matched regimes), the dominant terms give

$$T_{\text{think}} \approx 2 N_{\text{qTTT}} k,$$

which is Eq. equation 3.2. First-order corrections are $O\left(\frac{T_{\text{think}}}{T}\right)$ from the $\frac{T_{\text{think}}(T_{\text{think}}-1)}{2}$ term and $O\left(\frac{d}{T}\right)$ from $C_{\text{tok}}$.

**Sanity check (numeric instantiation).** Take $L=32$, $d=4096$, $r=4$ (a $\sim$7B dense model) and $T=10^5$. If the application budget allows decoding $T_{\text{think}}=8{,}000$ thinking tokens after prefill, the matched query-only schedules include, e.g., ($N_{\text{qTTT}}=10$, $k=400$) since $2 \cdot 10 \cdot 400 \approx 8{,}000$. This reallocation keeps the KV cache length fixed at $T$ and spends the same FLOPs to reshape queries against the existing $\{K, V\}$ instead of growing the cache with additional tokens.

## D  EXPERIMENTAL DETAILS

**Models and tokenization.** We evaluate Qwen3-{1.7B, 4B, 8B} with their native tokenizers and maximum supported context windows. All prompts use UTF-8, and inputs are delimited with explicit section headers (e.g., `[CONTEXT]`, `[QUESTION]`). Unless otherwise noted, we evaluate on the official validation/dev splits and follow each benchmark's scoring script.

**Decoding and "Thinking" budget.** We adopt model-recommended decoding parameters: *Thinking*: temperature=0.6, top-$p$=0.95, top-$k$=20; *Non-thinking*: temperature=0.7, top-$p$=0.8, top-$k$=20. We cap total generation length so that *Thinking* consumes exactly $T_{\text{think}}$ intermediate tokens plus the final answer; for compute matching, we use $T_{\text{think}} = 8192$ unless otherwise stated. Self-consistency/best-of-$n$ are *disabled* by default to keep FLOPs matched.

**Query-only TTT (query-only TTT) hyperparameters.** We update only $W_Q$ in all attention layers using AdamW (weight decay 0.01) with a sweep over learning rates $\{3e{-}4, 3e{-}5, 1e{-}5, 3e{-}6, 1e{-}6, 3e{-}7\}$; we report the best per-dataset LR selected on a held-out portion of the validation set. Batch size is 1 (long contexts). We perform $N_{\text{TTT}}$ span updates of length $k$ with a single prefill/cached $\{K, V\}$; unless stated otherwise, $(k, N_{\text{TTT}}) = (128, 32)$, compute-matched to *Thinking* via $T_{\text{think}} \approx 2N_{\text{TTT}}k$ (§3.3). Spans are sampled uniformly over $[1, T{-}k]$; gradient clipping at 1.0; bf16 precision. Additionally, we perform a sensitivity analysis of qTTT across learning rates. Table 1 shows the variation of accuracy on our synthetic tasks across context lengths. We find that qTTT is not very sensitive to the choice of LR: the performance is relatively consistent between $[1e{-}5, 1e{-}6]$ and only falls on the extreme values of LR.

Table 1: **Sensitivity to Learning Rate ($\eta$).** Performance of qTTT across varying learning rates. Extreme rates cause instability (high $\eta$) or insufficient adaptation (low $\eta$), with the optimal range typically between 1e-6 and 1e-5.

| Task / Context | 1e-4 | 3e-5 | 1e-5 | 3e-6 | 1e-6 | 3e-7 |
|---|---|---|---|---|---|---|
| *Bank Transactions* | | | | | | |
| 512 | 4.2 | 26.5 | **28.0** | 27.2 | 26.8 | 15.5 |
| 2,536 | 1.5 | 13.8 | **14.4** | 14.0 | 12.5 | 6.2 |
| 5,120 | 0.8 | **10.0** | 9.2 | 8.5 | 7.8 | 3.5 |
| 9,560 | 0.0 | 7.8 | **8.4** | 7.9 | 7.0 | 1.2 |
| *OLMo Code Bugs* | | | | | | |
| 512 | 8.5 | 42.0 | 44.5 | **45.7** | 43.2 | 22.0 |
| 2,050 | 5.1 | 38.5 | 40.2 | **41.6** | 39.5 | 18.5 |
| 7,450 | 2.2 | 25.0 | **28.0** | 27.5 | 24.8 | 10.5 |
| 10,000 | 1.0 | 18.2 | **20.2** | 19.5 | 17.8 | 5.2 |

**Evaluation metrics.** We use official scripts per subset: EM/F1 or dataset-specific accuracy for QA; ROUGE-{1,2,L} or benchmark-provided summary metrics for summarization; multiple-choice accuracy for `QAuLITY`. When a subset defines both EM and F1, we report the primary metric specified by the benchmark.

**Prompts and templates.** Below we provide the base non-thinking and thinking templates used per task family. All runs share the same template within a family across methods; *Thinking* adds a scratchpad section but the final answer must appear after a `Final:` tag.

*Non-thinking (base)*

```
[SYSTEM]
You are a careful assistant. Use only the provided context.
If the answer is not supported, output "unknown".
[TASK]
{TASK_DESCRIPTION}     # e.g., short answer QA / summary / MCQ
[CONTEXT]
{CONTEXT_BLOCKS}       # e.g., {DOCUMENTS}|{DIALOGUE}|{CODE}|{TABLE}
[QUESTION or INSTRUCTION]
{QUESTION_OR_INSTRUCTION}     # prompt for the required output
[CONSTRAINTS]
[ANSWER]
```

*Thinking (base)*

```
[SYSTEM]
Reason privately in [SCRATCHPAD],
then provide a single final output after "Final:".
If not supported by the context, output "Final: unknown".
[TASK]
{TASK_DESCRIPTION}
[CONTEXT]
{CONTEXT_BLOCKS}
[QUESTION or INSTRUCTION]
{QUESTION_OR_INSTRUCTION}
[SCRATCHPAD]
...     # hidden chain-of-thought tokens (capped to T_think)
[FINAL]
Final:
```

**Post-processing and extraction.** For "thinking" runs, we extract the substring after `Final:` (trim, strip quotes). For MCQ, we regex-match `[ABCD]`; for extractive QA, we normalize punctuation/whitespace (SQuAD-style). For summarization, we truncate to the requested budget (e.g., 200 words) and use the benchmark scorer verbatim.

**Compute matching and seeds.** Unless otherwise specified, *Thinking* uses $T_{\text{think}} = 8192$ and query-only TTT uses $(k, N_{\text{TTT}}) = (128, 32)$ so that $T_{\text{think}} \approx 2N_{\text{TTT}}k$. We fix the random seed for span sampling and decoding across methods per run; results are averaged over one run per configuration (low variance in our setting).

## E    SCORE DILUTION EVIDENCE ON LONG CONTEXTS

**Motivation.** Long-context failures could be a result of a multitude of reasons and design choices. Past literature in long-context modeling has primarily focused on tuning positional encoding to improve long-context abilities. Here we present some evidence supporting our claim that *score dilution* is one of the primary reasons for long-context failure. We show that as the context grows, attention mass on the target collapses, and accuracy falls even when rotary position embeddings (RoPE) are present and the model is not changed otherwise. We further show that qTTT counteracts this collapse suggesting that our approach actually counteracts score dilution in practice.

Table 2: Bank Transactions (Qwen3-4B): Accuracy (%) and attention mass vs. context length with and without RoPE, and with qTTT.

| Context Tokens | Thinking (RoPE) | | Thinking (No-RoPE) | | qTTT (Ours) | |
|---|---|---|---|---|---|---|
| | Acc | Mass | Acc | Mass | Acc | Mass |
| 512 | 36.00 | $0.46 \pm 0.04$ | 34.00 | $0.44 \pm 0.04$ | 28.00 | $0.42 \pm 0.06$ |
| 2,536 | 6.00 | $0.22 \pm 0.03$ | 5.00 | $0.20 \pm 0.02$ | 14.40 | $0.41 \pm 0.08$ |
| 5,120 | 2.50 | $0.11 \pm 0.02$ | 0.80 | $0.03 \pm 0.01$ | 10.00 | $0.36 \pm 0.09$ |
| 9,560 | 1.00 | $0.04 \pm 0.01$ | 0.50 | $0.01 \pm 0.00$ | 8.40 | $0.25 \pm 0.09$ |

Table 3: OLMo Code Bugs (Qwen3-4B): Accuracy (%) and attention mass vs. context length with and without RoPE, and with qTTT.

| Context Tokens | Thinking (RoPE) | | Thinking (No-RoPE) | | qTTT (Ours) | |
|---|---|---|---|---|---|---|
| | Acc | Mass | Acc | Mass | Acc | Mass |
| 512 | 50.00 | $0.64 \pm 0.05$ | 47.40 | $0.61 \pm 0.05$ | 45.70 | $0.58 \pm 0.06$ |
| 2,050 | 21.60 | $0.38 \pm 0.07$ | 16.20 | $0.29 \pm 0.04$ | 41.60 | $0.51 \pm 0.08$ |
| 7,450 | 17.20 | $0.26 \pm 0.06$ | 10.60 | $0.14 \pm 0.02$ | 28.00 | $0.42 \pm 0.09$ |
| 10,000 | 10.00 | $0.12 \pm 0.03$ | 3.00 | $0.04 \pm 0.01$ | 20.20 | $0.35 \pm 0.09$ |

**Experimental setting (RoPE ablation).**   We evaluate Qwen3-4B on two tasks (Bank Transactions; OLMo Code Bugs) under three test-time regimes: (1) *Thinking-only* with a fixed thinking budget (4k or 8k tokens), (2) *qTTT (ours)* with a brief query-only adaptation while reusing the prefetched KV cache, and (3) a *No-RoPE* ablation where we disable rotary phase application to $Q/K$ at inference (identity rotation), keeping all weights, prompts, and budgets unchanged and without any additional fine-tuning. This isolates the role of positional encoding while holding training and data fixed.

**Attention-mass metric.**   For each decode step $t$, layer $\ell$, and head $h$, let $A_{t,\tau}^{(\ell,h)}$ denote the softmax attention from the current query to context position $\tau$. Given a labeled set of target indices $\mathcal{T}$, we define the *attention mass* at step $t$ as $\sum_{\tau \in \mathcal{T}} A_{t,\tau}^{(\ell,h)}$, then average over all layers and heads; for multi-token answers we average over their output steps. We report mean $\pm$ std across multiple runs.

**Findings.**   Tables 2 and 3 show that thinking-only accuracy and attention mass both decay sharply with context length. Disabling RoPE accelerates this collapse (lower mass and accuracy), but *even with* RoPE the decline is substantial. In contrast, qTTT sustains markedly higher attention mass as context grows and correspondingly improves accuracy. These results support the view that score dilution, rather than training-data scarcity alone, is the dominant failure mode in these settings.

## F    ZeroScrolls and LongBench-v2: All models and subsets.

This appendix reports the complete breakdowns for all benchmarks, models, and inference settings. We compare three modes—vanilla in-context, chain-of-thought "Thinking", and our test-time training method (qTTT)—for Qwen3-1.7B/4B/8B across LongBench-v2 and ZeroScrolls. Unless otherwise noted, higher is better and bold indicates the best within each row/condition.

Figure 8 shows a FLOP-matched overview of LongBench-v2 results across its six domains. The detailed per-domain numbers that underlie this figure appear in Table 4. Figure 9 summarizes the observed results on ZeroScrolls. The complete per-dataset numbers, including retrieval-heavy and summarization tasks, are provided in Table 5.

## G    Additional Test-Time Scaling Baselines

**Baselines.**   We compare **Best-of-$N$ (BoN)** and **Beam Search** to our method under strict compute parity. *BoN / Self-Consistency (SC-$N$):* we run $N$ independent decodes, each with an equal share of

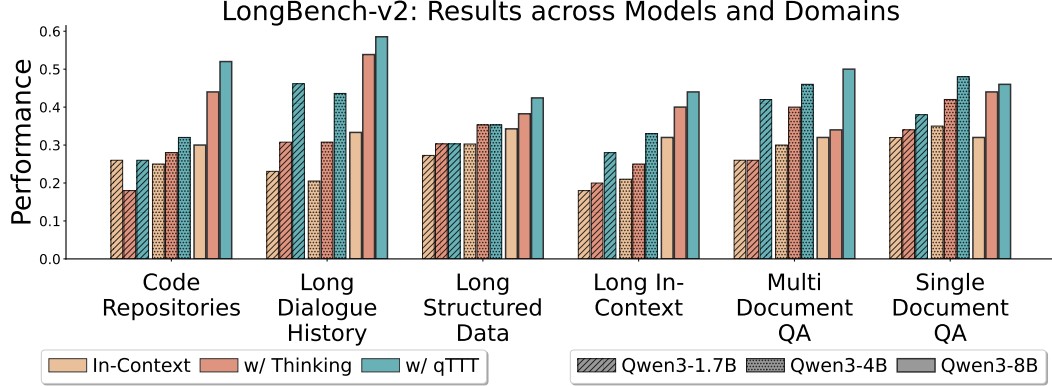

Figure 8: FLOP-matched comparison on **LongBench-v2** (Bai et al., 2023b) across six domains for Qwen3-1.7B/4B/8B under vanilla in-context only, with thinking (CoT), and with test-time training (TTT). TTT consistently yields the best accuracy across domains and model sizes, with the largest gains on long-dialogue and document-QA tasks, and benefits growing with model size.

Table 4: Full **LongBench-v2** results for Qwen3-1.7B/4B/8B under In-context, Thinking, and qTTT. Scores follow benchmark-defined metrics; bold marks the best within each row/condition.

| | Qwen3-1.7B | | | Qwen3-4B | | | Qwen3-8B | | |
|---|---|---|---|---|---|---|---|---|---|
| | **In-context** | **Thinking** | **qTTT** | **In-context** | **Thinking** | **qTTT** | **In-context** | **Thinking** | **qTTT** |
| Code Repositories | **26.0** | 18.0 | **26.0** | 25.0 | 28.0 | **32.0** | 30.0 | 44.0 | **52.0** |
| Long Dialogue History | 23.1 | 30.8 | **46.2** | 20.5 | 30.8 | **43.6** | 33.3 | 53.8 | **58.5** |
| Long Structured Data | 27.3 | **30.3** | **30.3** | 30.3 | **35.3** | **35.3** | 34.3 | 38.2 | **42.4** |
| Long In-Context | 18.0 | 20.0 | **28.0** | 21.0 | 25.0 | **33.0** | 32.0 | 40.0 | **44.0** |
| Multi-Document QA | 26.0 | 26.0 | **42.0** | 30.0 | 40.0 | **46.0** | 32.0 | 34.0 | **50.0** |
| Single-Document QA | 32.0 | 34.0 | **38.0** | 35.0 | 42.0 | **48.0** | 32.0 | 44.0 | **46.0** |
| **Average** | 25.4 | 26.5 | **35.1** | 27.0 | 33.5 | **39.6** | 32.3 | 42.3 | **48.8** |

the extra reasoning budget, and select the final answer by majority vote (ties broken by sequence log-prob). *Beam-$k$:* we run left-to-right beam search of width $k$; to enforce parity with other test-time scaling, the *total* added "thinking" tokens across all beams is fixed.

**Design choices (strict matching).** We match all methods to a fixed extra budget corresponding to $T_{\text{think}} = 8192$ tokens beyond the vanilla decode. SC-$N$ allocates $\approx 8192/N$ tokens to each sample; Beam-$k$ allocates $\approx 8192/k$ tokens per beam. All results use the same prompt, output length (128 tokens); latencies are reported separately in §H. This protocol removes budget-induced confounders and isolates the effect of test-time scaling itself.

**Conclusion.** Across both LongBench-v2 and ZeroScrolls (Qwen3-4B), qTTT is competitive with or better than strictly FLOP-matched BoN and Beam. SC-$N$ helps when single-run accuracy is already high (e.g., *Single Document QA*, *QUALITY*), but often degrades when the per-sample accuracy is below 50%. Beam-$k$ provides only modest gains under equal budgets due to correlated beams and imperfect ranking, and frequently trails qTTT.

## H  LATENCY AND COMPUTE-MATCHED MEASUREMENTS

**Setup.** All latency numbers were measured on a single NVIDIA A100 GPU in standard inference mode. We report wall-clock time in seconds (mean ± std) for three different context lengths. For a given model size and context length, we perform latency analysis based on the amount of FLOPs, $F_{qTTT}$, it takes to run $N_{qTTT} = 32$ steps for $k = 128$ on a single evaluation example. We report the following metrics:

- $N_{\text{think}}$: Number of thinking tokens that can be generated to match $F_{qTTT}$ FLOPs.
- $N_{\text{BoN}}$: Number of best-of-N trajectories that can be generated to match $F_{qTTT}$ FLOPs.

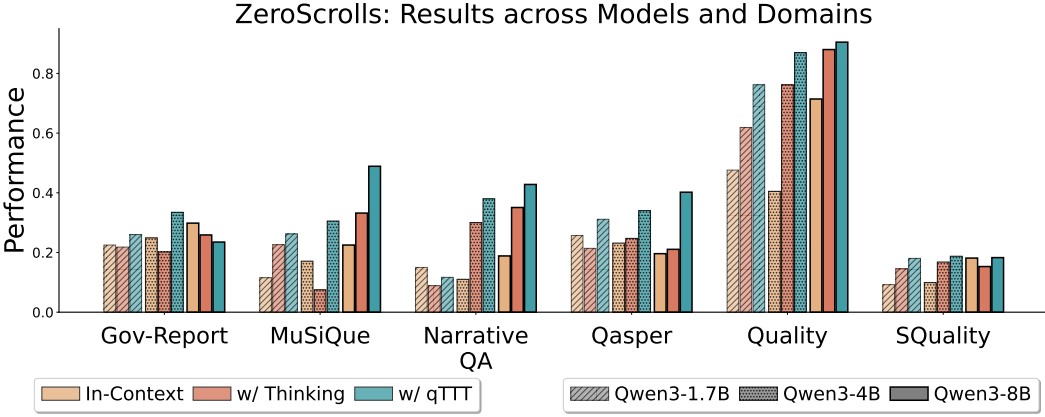

Figure 9: FLOP-matched comparison on the **ZeroScrolls** benchmark (Shaham et al., 2023) for Qwen3-1.7B/4B/8B under long contexts, with thinking (CoT), and with test-time training (TTT). TTT achieves the highest scores on nearly all datasets—especially on the retrieval-focused tasks, with a general increase with model size.

Table 5: Full **ZeroScrolls** results across eight datasets for Qwen3-1.7B/4B/8B under In-context, Thinking, and qTTT. Datasets span retrieval and summarization; bold marks the best within each row/condition (higher is better).

| | Qwen3-1.7B | | | Qwen3-4B | | | Qwen3-8B | | |
|---|---|---|---|---|---|---|---|---|---|
| | In-context | Thinking | qTTT | In-context | Thinking | qTTT | In-context | Thinking | qTTT |
| GovReport | 22.5 | 21.8 | **26.0** | 24.9 | 20.2 | **33.5** | 22.0 | 17.8 | **29.8** |
| MuSiQue | 11.6 | 22.6 | **26.2** | 17.1 | 7.5 | **30.5** | 22.5 | 43.9 | **48.9** |
| NarrativeQA | 15.0 | 8.9 | **11.7** | 11.0 | 30.0 | **38.0** | 18.9 | 35.1 | **42.8** |
| QASPER | 25.7 | 21.4 | **31.1** | 23.2 | 24.7 | **34.0** | 19.6 | 21.1 | **26.1** |
| QMSum | 6.2 | 7.5 | **9.5** | **10.9** | 7.7 | 8.6 | **9.8** | 8.6 | 8.6 |
| QUALITY | 47.6 | 61.9 | **76.2** | 40.5 | 76.2 | **87.0** | 71.4 | 90.5 | **94.5** |
| SQuALITY | 9.2 | 14.6 | **18.0** | 9.9 | 16.8 | **18.7** | 18.1 | 15.3 | **18.3** |
| SummScreen-FD | 8.2 | 7.2 | **7.4** | **9.9** | 8.3 | **9.9** | **10.4** | 7.9 | 7.9 |
| **Average** | 18.3 | 20.7 | **25.8** | 18.4 | 23.9 | **32.5** | 24.1 | 30.0 | **34.6** |

- $t_{\text{ICL}}$: Wall-clock time for a vanilla in-context pass on single example. This roughly corresponds to the prefill time.

- $t_{\text{think}}$: Wall-clock time to generate $N_{\text{think}}$ tokens, given a single example.

- $t_{\text{BoN}}$: The amount of time to compute best-of-N via self-consistency for $N_{\text{BoN}}$ trajectories given a single example.

- $t_{\text{qTTT}}$: The amount of time to perform $N_{qTTT} = 32$ steps of qTTT steps with span length $k = 128$ for a single example.

Tables 10, 11, 12 show the results of the measurements on Qwen3-1.7B, 4B, and 8B, respectively. We find that the wall-clock time for all three test-time compute strategies—qTTT, thinking, and best-of-N—is quite similar. We also note that prefilling the KV cache, which is approximately equal to $t_{\text{ICL}}$ dominates most of the decoding time, especially for longer sequence lengths. This motivates the frozen K/V attention weights in our setup, without which the prefill would need to be recomputed with every training step.

Table 6: **Qwen3-32B on LongBench-v2.** Comparison of In-context, Thinking, and qTTT. These findings demonstrate that that the improvements with qTTT hold across model scales.

|  | In-context | Thinking | qTTT |
|---|---|---|---|
| Code Repositories | 36.00 | 61.00 | **74.00** |
| Long In-Context | 44.00 | 56.00 | **57.00** |
| Long Structured Data | 39.30 | 42.20 | **51.50** |
| Long Dialogue History | 47.10 | **77.90** | 75.50 |
| Multi Document QA | 35.00 | 41.00 | **56.00** |
| Single Document QA | 36.00 | 47.00 | **49.00** |

Table 7: **Qwen3-32B on ZeroScrolls.** Comparison of In-context, Thinking, and qTTT. These findings demonstrate that that the improvements with qTTT hold across model scales.

|  | In-context | Thinking | qTTT |
|---|---|---|---|
| Gov Report | **26.70** | 24.80 | 26.00 |
| Musique | 28.90 | 54.90 | **59.20** |
| Narrative QA | 27.70 | 42.40 | **49.60** |
| Qasper | 24.10 | 35.00 | **42.40** |
| QMSum | **11.90** | 9.90 | 10.80 |

Table 8: LongBench-v2 (Qwen3-4B): Strict FLOP-matched test-time scaling. Numbers are accuracies (%). SC-$N$ uses $8192/N$ tokens per sample; Beam-$k$ uses $8192/k$ tokens per beam.

| Task | Thinking | qTTT | SC-8 | SC-16 | SC-32 | Beam-8 | Beam-16 | Beam-32 |
|---|---|---|---|---|---|---|---|---|
| Code Repositories | 28.0 | 32.0 | 30.5 | 18.4 | 5.2 | 27.5 | 15.1 | 4.8 |
| Long In-Context | 25.0 | 33.0 | 28.5 | 20.1 | 8.5 | 26.0 | 18.5 | 7.2 |
| Long Structured Data | 35.3 | 35.3 | 36.1 | 30.5 | 12.2 | 34.8 | 28.1 | 11.0 |
| Long Dialogue History | 30.8 | 43.6 | 34.2 | 31.0 | 15.5 | 29.5 | 25.2 | 12.0 |
| Multi Document QA | 40.0 | 46.0 | 44.5 | 41.2 | 25.8 | 39.8 | 35.5 | 22.1 |
| Single Document QA | 42.0 | 48.0 | 45.5 | 49.2 | 51.8 | 43.5 | 44.2 | 41.0 |
| **Avg.** | 33.5 | **39.7** | 36.6 | 31.7 | 19.8 | 33.5 | 27.8 | 16.4 |

Table 9: ZeroScrolls (Qwen3-4B): Strict FLOP-matched test-time scaling. Numbers are accuracies (%). SC-$N$ uses $8192/N$ tokens per sample; Beam-$k$ uses $8192/k$ tokens per beam.

| Task | Thinking | qTTT | SC-8 | SC-16 | SC-32 | Beam-8 | Beam-16 | Beam-32 |
|---|---|---|---|---|---|---|---|---|
| gov report | 20.2 | 33.5 | 24.5 | 15.2 | 2.1 | 22.8 | 12.5 | 1.8 |
| musique | 7.5 | 30.5 | 18.2 | 12.5 | 4.5 | 14.5 | 9.8 | 3.2 |
| narrative qa | 30.0 | 38.0 | 35.5 | 32.0 | 22.5 | 31.2 | 25.5 | 18.1 |
| qasper | 24.7 | 34.0 | 28.5 | 22.1 | 10.5 | 26.5 | 19.5 | 9.2 |
| qmsum | 7.7 | 8.6 | 9.2 | 5.1 | 0.8 | 8.5 | 4.5 | 0.7 |
| quality | 76.2 | 87.0 | 82.5 | 85.1 | 84.5 | 78.5 | 76.2 | 70.1 |
| squality | 16.8 | 18.7 | 17.5 | 19.2 | 20.5 | 17.1 | 17.5 | 17.8 |
| summ screen fd | 8.3 | 9.9 | 9.5 | 6.5 | 1.2 | 8.8 | 5.5 | 1.1 |
| **Avg.** | 23.9 | **32.5** | 28.2 | 24.7 | 18.3 | 26.0 | 21.4 | 15.3 |

Table 10: Latency and wall-clock time comparisons given a fixed FLOP budget for Qwen3-1.7B.

| Context Length | $t_{\text{ICL}}$ (s) | $t_{\text{qTTT}}$ (s) | $t_{\text{think}}$ (s) | $t_{\text{BoN}}$ (s) | $N_{\text{think}}$ | $N_{\text{BoN}}$ |
|---|---|---|---|---|---|---|
| 8,000 | $8.73 \pm 0.35$ | $16.92 \pm 0.68$ | $16.93 \pm 0.68$ | $16.05 \pm 0.64$ | 1,434 | 11 |
| 32,000 | $34.93 \pm 1.40$ | $43.12 \pm 1.72$ | $43.11 \pm 1.72$ | $40.78 \pm 1.63$ | 358 | 3 |
| 128,000 | $139.70 \pm 5.59$ | $147.89 \pm 5.92$ | $147.93 \pm 5.92$ | $139.70 \pm 5.59$ | 90 | 1 |

Table 11: Latency and wall-clock time comparisons given a fixed FLOP budget for Qwen3-4B.

| Context Length | $t_{\text{ICL}}$ (s) | $t_{\text{qTTT}}$ (s) | $t_{\text{think}}$ (s) | $t_{\text{BoN}}$ (s) | $N_{\text{think}}$ | $N_{\text{BoN}}$ |
|---|---|---|---|---|---|---|
| 8,000 | $14.61 \pm 0.58$ | $28.27 \pm 1.13$ | $28.26 \pm 1.13$ | $27.41 \pm 1.10$ | 1,365 | 11 |
| 32,000 | $58.45 \pm 2.34$ | $72.11 \pm 2.88$ | $72.09 \pm 2.88$ | $68.69 \pm 2.75$ | 341 | 3 |
| 128,000 | $233.81 \pm 9.35$ | $247.47 \pm 9.90$ | $247.41 \pm 9.90$ | $233.81 \pm 9.35$ | 85 | 1 |

Table 12: Latency and wall-clock time comparisons given a fixed FLOP budget for Qwen3-8B.

| Context Length | $t_{\text{ICL}}$ (s) | $t_{\text{qTTT}}$ (s) | $t_{\text{think}}$ (s) | $t_{\text{BoN}}$ (s) | $N_{\text{think}}$ | $N_{\text{BoN}}$ |
|---|---|---|---|---|---|---|
| 8,000 | $22.13 \pm 0.89$ | $42.61 \pm 1.70$ | $42.62 \pm 1.70$ | $41.33 \pm 1.65$ | 1,229 | 10 |
| 32,000 | $88.53 \pm 3.54$ | $109.01 \pm 4.36$ | $109.00 \pm 4.36$ | $97.07 \pm 3.88$ | 307 | 2 |
| 128,000 | $354.13 \pm 14.17$ | $374.61 \pm 14.98$ | $374.67 \pm 14.99$ | $354.13 \pm 14.17$ | 77 | 1 |