# OpenReview forum: "Let's (not) just put things in Context: Test-time Training for Long-context LLMs"
_ICLR.cc/2026/Conference — ICLR 2026 Poster_

### Official Review · Reviewer_SSXV · 2025-10-28

**Soundness:** 2
**Presentation:** 3
**Contribution:** 2
**Rating:** 2
**Confidence:** 4

**Summary:**

The paper argues that long-context failures in static self-attention stem from score dilution: as distractors accumulate, attention mass on the true “needle” vanishes unless the target–distractor logit gap grows with context length. The authors prove a logarithmic margin requirement and show that decoding-based inference-time scaling (e.g., “thinking” tokens) cannot reliably fix this. They propose query-only test-time training (qTTT): perform a single prefill to cache K/V, then run a few lightweight gradient steps updating only the query projections over short spans while reusing the KV cache.

**Strengths:**

1. The paper proposes a compute-aware design. The proposed prefill-once, KV-cache reuse, and FLOP matching to thinking tokens make for a fair comparison and a practical recipe.

2. The benchmarks in the paper are sufficient. It evaluates the proposed method across model sizes and multiple long-context benchmarks.

3. The idea is really cute. The inference-time updating the parameters is novel in the community.

**Weaknesses:**

1. The theoretical analysis is too naive to capture the main motivation. Concretely, it cannot prove that the score dilution is the main reason for the poor performance.

* For example, whether the poor performance comes from the small number of training samples. Usually, learning more complex abilities, i.e., solving problems with longer context, requires more training samples than learning the simple ability. The poor performance can simply come from the relatively small number of samples.

* Even we assume that the poor performance comes from the score dilution. The current analysis is oversimplified. For example, Lemma 2.3 requires a $log T$ scaling of the logits. Such can be achieved by RoPE, which is designed for the long-range decay. In addition, adapting RoPE for the proper score decaying is the cornerstone of most long-context papers. The existing analysis just ignores the role of RoPE.

2. The paper claims that the proposed method solves the score dilution problem. However, no evaluation or visualization of the attention scores are presented.

3. Test-time scaling baseline is missing. The proposed method achieves better performance with more computation. However, no such baseline is included. For example, whether the beam search, BoN achieves the comparable performance with the same budget.

4. Ablation is missing. The learning rate for the test-time training is important. Whether this depends on the context length, context semantic is not known.

**Questions:**

See the weaknesses.

---

> ### Author Response · Authors · 2025-11-20
> **Authors Response (1/2)**
>
> We thank the reviewer for their careful analysis of our work. We appreciate the reviewer’s acknowledgment of the novelty of our approach, compute-aware design, and thoroughness of our empirical evaluations. We address the reviewers’ concerns below:
>
> **Is score dilution really the reason for poor long-context performance? Does qTTT solve score dilution?**
>
> The reviewer makes an important point that poor long-context performance can stem from various factors including the choice of position encoding and lack of long training sequences. We acknowledge that these are important factors for long-context performance. However, our paper claims that score dilution is *one of the key* factors and it motivates our test-time training approach. Moreover, we show that our approach helps empirically across a variety of settings.
>
> Based on the reviewer’s feedback, we conduct further empirical analysis to understand the effect of score dilution in contrast to other factors. We conduct a thorough study **analyzing attention mass** over target tokens. We find that poor long-context performance persists with relative position encodings as well as settings where enough training samples exist. In all these cases, we find that **qTTT continues to help**. We enlist our key findings here:
>
> - *Role of RoPE*
>
> We conduct additional **ablations with and without RoPE** on Qwen3 models. We find that using RoPE indeed helps with long context scenarios. However, we still see significant reduction in performance and attention mass on the target tokens when the context length is increased, both with and without RoPE. We claim that this can largely be attributed to score dilution. Moreover, we find that qTTT helps preserve performance and attention mass to a large extent. We add a complete set of these results in **Appendix E (Tables 2 and 3)** and include a concise summary here:
>
> | #Input tokens | Thinking (RoPE) |           | Thinking (no RoPE) |           | qTTT (RoPE) |           |
> |-----------------|--------------------|-----------|-----------------------|-----------|-------------|-----------|
> |                 | Accuracy                | Attention Mass on Target      | Accuracy                   |  Attention Mass on Target      | Accuracy         |  Attention Mass on Target      |
> |             512 |             50.0 | 0.64 |                47.4 | 0.61 |      45.7 | 0.58 |
> |           2,050 |             21.6 | 0.38 |                16.2 | 0.29 |      41.6 | 0.518 |
> |           7,450 |             17.2 | 0.26 |                10.6 | 0.14 |      28.0 | 0.42 |
> |          10,000 |             10.0 | 0.12 |                 3.0 | 0.04 |      20.2 | 0.35 |
>
> Note that this finding is consistent with prior work that shows, across a variety of synthetic settings, that a variety of relative positional encoding methods still suffer from phenomena like score dilution and continue to show poor performance [Figure 4 in Liu et. al., 2023].
>
> [Liu et. al., 2023]: Exposing Attention Glitches with Flip-Flop Language Modeling; Bingbin Liu, Jordan T. Ash, Surbhi Goel, Akshay Krishnamurthy, Cyril Zhang
>
> - *Influence of limited long sequences during training*
>
> We agree with the reviewer that training on more long sequences would help improve a lot of long context issues. However, in this work we are working with pre-trained models with limited control on the training data. We believe it’s still worthwhile to investigate approaches to improve length generalization without having to train on extremely long sequences.
>
> We perform an additional analysis to investigate the effectiveness of qTTT when sufficient training examples exist. Since Qwen3 models have been trained on up to 32k tokens, we **sub-sample examples** from the ZeroScrolls benchmark where the input context has **5k to 10k length**. We perform evaluations on these examples and find that poor performance still holds on this set of examples for vanilla ICL and thinking baselines. We note that **qTTT still helps** in these settings, pointing to the potential overhead in long-context performance and effectiveness of qTTT in achieving this.
>
> | Avg #tokens = 7.3k | In-Context | Thinking | qTTT |
> |--------------------|------------|-------------|---------|
> | Qwen3-1.7B         |       18.7 |        22.4 |    27.8 |
> | Qwen3-4B           |       19.3 |        23.1 |    31.7 |
> | Qwen3-8B           |       24.8 |        29.3 |    33.4 |
>
> Note that these evaluations are performed with RoPE enabled. These results suggest that despite significant training, models still suffer on medium/long context tasks, simply as a result of having a lot of tokens (“distractors”) in context leading to score dilution.

---

> ### Author Response · Authors · 2025-11-20
> **Authors Response (2/2)**
>
> - *qTTT preserves attention mass on target tokens*
>
> Based on the reviewer’s suggestion, we conduct a thorough **analysis of attention mass** on our synthetic tasks. As reported in the first table above, qTTT greatly preserves the attention mass with an increasing number of context tokens. This is a direct indicator of preventing score dilution. We report a more detailed analysis, including confidence bounds and results on other tasks in **Appendix E (Tables 2 and 3)** of our paper. Overall, we find that qTTT consistently preserves attention mass and leads to better performance.
>
> **Additional test-time scaling baselines**
>
> We have now added additional results performing FLOP-matched comparisons with **best-of-N** and **beam search**. For both, we comprehensively evaluate different choices of “N” and number of beams. We include the full results in **Appendix G (Tables 8 and 9)**. The following table summarizes the key results with the best performance for best-of-N and beam search:
>
> | Qwen3-4B          | Thinking  | Best-of-N | Beam Search | qTTT |
> |-------------------|-----------|-----------|-------------|------|
> | Avg. LongBench-v2 | 33.5      | 36.6      | 33.5        | 39.7 |
> | Avg. ZeroScrolls  | 23.9      | 28.2      | 26.0        | 32.5 |
>
>
> We also perform additional evaluations on a larger model, **Qwen-32B**, and show that our findings extend to such large model scales. We report the full results across all subsets of LongBench-v2 and ZeroScrolls in **Appendix G (Tables 6 and 7)**. Here is a concise summary of the results:
>
> | Qwen3-32B         | In-context | Thinking |  qTTT |
> | :---------------- | --------------: | -------: | ----: |
> | Avg. LongBench-v2 |           39.57 |    54.18 | 60.50 |
> | Avg. ZeroScrolls  |           23.86 |    33.40 | 37.60 |
>
> **Sensitivity to learning rate**
>
> We have now added a sensitivity analysis across learning rates in **Appendix D (Table 1)**.
> We examine learning rates in the range [1e-4, 3e-7] for our synthetic tasks across various context lengths. We find that qTTT is **not** very sensitive to the choice of LR: the performance is relatively consistent between [3e-5, 1e-6] and only falls on the extreme values.
> Here is a concise summary of the findings for the reviewer’s reference:
>
> | LR                                                   | 1.00E-04 | 3.00E-05 | 1.00E-05 | 3.00E-06 | 1.00E-06 | 3.00E-07 |
> | :--------------------------------------------------- | -------: | -------: | -------: | -------: | -------: | -------: |
> | Avg. accuracy for error logs task |     1.63 |    14.53 |    15.00 |    14.40 |    13.53 |     6.60 |
> | Avg. accuracy for OLMo bugs task   |     4.20 |    30.93 |    33.23 |    33.58 |    31.33 |    14.05 |

---

### Official Review · Reviewer_Yzb3 · 2025-10-31

**Soundness:** 2
**Presentation:** 2
**Contribution:** 2
**Rating:** 4
**Confidence:** 3

**Summary:**

This paper addresses a critical limitation of long-context Large Language Models (LLMs): while modern LLMs support context windows of millions of tokens, they often fail to reliably use information buried in long texts. Existing inference-time strategies (e.g., chain-of-thought "thinking tokens") show diminishing returns as context length grows, due to a phenomenon the authors term score dilution—static self-attention cannot sufficiently separate the "target" (relevant information) from "distractor" (irrelevant) tokens, leading to vanishing target probability.

**Strengths:**

*  It formally introduces "score dilution" to explain long-context LLM failures, turning vague issues (e.g., missed key info) into a quantifiable, solvable problem—filling a gap in prior research that lacked clear theoretical grounding for such limitations.

* The proposed query-only Test-Time Training (qTTT) is innovative in its frugality: it reuses frozen KV caches and only updates query projections, avoiding the high compute of full-model fine-tuning or ineffective "thinking tokens" for long texts.

* qTTT offers a low-overhead, drop-in fix for real-world long-context use cases (code analysis, EHR review), and its "score dilution" framework guides future research on improving long-context LLMs beyond incremental tweaks.

**Weaknesses:**

* While it highlights compute efficiency, it does not measure inference latency (critical for production) when qTTT is added—leaving unclear if its small compute overhead translates to acceptable delays for time-sensitive tasks (e.g., real-time code debugging).

* It does not explore how qTTT performs with noisy or low-quality long texts (e.g., unstructured logs, messy code), where distractors are more prevalent—limiting understanding of its robustness beyond clean benchmark datasets.

**Questions:**

Does this paper provide a performance comparison with alternative test-time scaling methods, given an equivalent computational budget?

---

> ### Author Response · Authors · 2025-11-20
> **Authors Response**
>
> We thank the reviewer for their valuable feedback. We are glad the reviewer appreciates our theoretical formalization of long-context failure modes and also our proposed method.
>
> **Additional test-time scaling baselines**
>
> Based on the reviewer’s feedback, we added additional results performing FLOP-matched comparisons with **best-of-N** and **beam search**. For both, we comprehensively evaluate different choices of “N” and number of beams. We include the full results in **Appendix G (Tables 8 and 9)**. The following table summarizes the key results with the best performance for best-of-N and beam search:
>
> | Qwen3-4B          | Thinking  | Best-of-N | Beam Search | qTTT |
> |-------------------|-----------|-----------|-------------|------|
> | Avg. LongBench-v2 | 33.5      | 36.6      | 33.5        | 39.7 |
> | Avg. ZeroScrolls  | 23.9      | 28.2      | 26.0        | 32.5 |
>
> We also perform additional evaluations on a larger model, **Qwen-32B**, and show that our findings extend to these large model scales. We report the full results across all subsets of LongBench-v2 and ZeroScrolls in **Appendix G (Tables 6 and 7)**. Here is a concise summary of the results:
>
> | Qwen3-32B         | In-context only | Thinking |  qTTT |
> | :---------------- | --------------: | -------: | ----: |
> | Avg. LongBench-v2 |           39.57 |    54.18 | 60.50 |
> | Avg. ZeroScrolls  |           23.86 |    33.40 | 37.60 |
>
>
> **Inference latency**
>
> We conducted a comprehensive inference latency and wall-clock time analysis. We include the full measurements in **Appendix H (Tables 10 and 11)**. For the reviewer’s reference, here are some key numbers for the Qwen3-4B and Qwen3-8B models, respectively:
>
> | #Input tokens |  Prefill (s)  |    qTTT (s)   | Thinking (s) | Best-of-N (s) |
> |:--------------:|:-------------:|:-------------:|:-------------------:|:-------------:|
> | 32,000         | 58.45 ± 2.34  | 72.11 ± 2.88  | 72.09 ± 2.88        | 68.69 ± 2.75  |
> | 128,000        | 233.81 ± 9.35 | 247.47 ± 9.90 | 247.41 ± 9.90       | 233.81 ± 9.35 |
>
>
> | #Input tokens |   Prefill (s)  |    qTTT (s)    | Thinking (s) |  Best-of-N (s) |
> |:--------------:|:--------------:|:--------------:|:-------------------:|:--------------:|
> | 32,000         | 88.53 ± 3.54   | 109.01 ± 4.36  | 109.00 ± 4.36       | 97.07 ± 3.88   |
> | 128,000        | 354.13 ± 14.17 | 374.61 ± 14.98 | 374.67 ± 14.99      | 354.13 ± 14.1
>
> We find that the wall-clock time for all three test-time compute strategies, qTTT, thinking, and best-of-N, is very close. We also note that prefilling the KV cache dominates most of the decoding time, especially for longer sequence lengths. This motivates our approach of keeping the K/V weights frozen, without which the prefill would need to be recomputed after every training step.
>
> **Including settings with noisy or low-quality long texts**
>
> We note that our evaluation benchmarks, specifically ZeroScrolls and LongBench-v2, were selected precisely because they contain significant unstructured noise and deliberate distractors, and go beyond clean long-context settings.
>
> - **QMSum**: Our results on QMSum (part of ZeroScrolls) directly demonstrate performance on "unstructured logs." QMSum consists of meeting transcripts filled with spoken disfluencies, interruptions, and non-linear dialogue history. As noted by **Shaham et al. (2023)**, this task specifically penalizes models that cannot filter out the "conversational noise" to track the main narrative.
>
> - **Multi-Document QA**: The Multi-Document QA and Musique tasks in our evaluation are inherently adversarial. These datasets require the model to identify a "needle" of reasoning amidst a large "haystack" of irrelevant documents (distractors). **Bai et al., 2023** explicitly characterizes these tasks as measuring robustness against "noisy contexts" where relevant information is diluted.
>
> - **Code Repositories**: The Code Repositories task (from LCC/RepoBench) utilizes real-world GitHub repositories, which naturally include "messy" elements such as commented-out blocks and noisy logs.
>
> Performance improvement with qTTT across all these subsets indicate its robustness to unstructured and noisy input. We now also include an analysis of attention mass in **Appendix E (Tables 2 and 3)** that further supports that qTTT maintains high attention weights on relevant tokens even when the context length (and thus the noise/distractor ratio) increases.

---

> > ### Comment · Reviewer_Yzb3 · 2025-11-27
> > **Official Comment by Reviewer  Yzb3**
> >
> > Thank you for the rebuttal.
> >
> > I would suggest the authors compare their method against more advanced (or state-of-the-art) test-time scaling methods under an equivalent computational budget. If the proposed method demonstrates competitive performance in such a comparison, I would be happy to raise my score.

---

> > > ### Author Response · Authors · 2025-11-27
> > >
> > > We thank the reviewer for their response and the openness to raising their score.
> > >
> > > We would like to highlight that our rebuttal results already include comparisons against the strongest standard test-time scaling strategies, strictly controlled for computational budget. Specifically, we compare against Best-of-N with Self-Consistency and Beam Search.
> > >
> > > We implement Best-of-N as Self-Consistency, where we sample N independent reasoning paths and aggregate the final answer via majority vote. This is widely recognized as the dominant strategy for scaling test-time compute. Recent work on inference scaling laws (Snell et al., 2024) and reasoning robustness (Huang et al., 2024) identifies parallel sampling (Best-of-N) as the primary, high-performance baseline that is notoriously difficult to beat without external oracles.
> > >
> > > As shown in Appendix G, qTTT significantly outperforms this "hard-to-beat" baseline when given the exact same compute budget:
> > >
> > > | Method (FLOP-Matched) | Avg. LongBench-v2 | Avg. ZeroScrolls |
> > > | :--- | :---: | :---: |
> > > | Thinking Tokens (Standard Scaling) | 33.5 | 23.9 |
> > > | Beam Search | 33.5 | 26.0 |
> > > | Best-of-N + Self-Consistency | 36.6 | 28.2 |
> > > | **qTTT (Ours)** | **39.7** | **32.5** |
> > >
> > >
> > > **Request for Clarification:** Given that we outperform Self-Consistency and Beam Search, could you kindly specify which other specific test-time scaling methods you have in mind?
> > >
> > > We note that methods like Process Reward Models (PRMs) require training external verifiers on separate datasets, and Self-Critique generally emerges reliably only at very large model scales (Lin et al., 2024) or often degrades performance compared to self-consistency (Huang et al., 2024). Since qTTT is a general-purpose strategy that requires no external data or task-specific adaptation, we believe Best-of-N and Beam Search are the most rigorous direct comparisons. However, if there are other general-purpose methods suitable for this setting, we would be happy to consider them.
> > >
> > > **References:**
> > >
> > > [Snell et al., 2024] *Scaling LLM Test-Time Compute Optimally can be More Effective than Scaling Model Parameters*
> > >
> > > [Huang et al., 2024] *Large Language Models Cannot Self-Correct Reasoning Yet*
> > >
> > > [Lin et al., 2024] *CriticBench: Benchmarking LLMs for Critique-Correct Reasoning*

---

### Official Review · Reviewer_BjzK · 2025-10-31

**Soundness:** 3
**Presentation:** 4
**Contribution:** 3
**Rating:** 6
**Confidence:** 3

**Summary:**

The author proposes a test-time learning method for long context handling with ICL examples.

**Strengths:**

- Clever idea to learn only the test time "decoder", not the "encoder"
- Extremely strong performance improvement

**Weaknesses:**

- Training required (during decoding)
  - No detailed efficiency study has happened.
- No large model is tested

**Questions:**

- What if training the whole model from scratch using this method? (Including encoder, meta learning approach)
- Why do you need to update the query parameters? No need to finetune the MLP?
- How can we serve different query weight parameters in a real-world serving framework, such as vLLM?
  - This could be challenging due to the CUDA graph capturing.

---

> ### Author Response · Authors · 2025-11-20
> **Authors Response**
>
> We thank the reviewer for their feedback. We acknowledge the reviewer’s appreciation of our idea and the significance of the performance gains we see. We address the reviewer’s concerns and questions below:
>
> **Evaluations on larger models**
>
> Based on the reviewer’s feedback, we now include results on the **Qwen3-32B** model as a proxy for a large model. We report the full results across all subsets of LongBench-v2 and ZeroScrolls in **Appendix G (Tables 6 and 7)**. We include a concise summary of the results here for a quick reference:
>
> | Qwen3-32B         | In-context only | Thinking |  qTTT |
> | :---------------- | --------------: | -------: | ----: |
> | Avg. LongBench-v2 |           39.57 |    54.18 | 60.50 |
> | Avg. ZeroScrolls  |           23.86 |    33.40 | 37.60 |
>
> Overall, we find that the improvements via qTTT hold consistently on the 32B model across subsets of both LongBench-v2 and ZeroScrolls.
>
>
> **What if we train the whole model? Why do we only train the query vectors?**
>
> As a clarification to the reviewer, our setting uses decoder-only LLMs and no separate encoder. We restrict ourselves to training the query matrices since full-parameter test-time training would invalidate KV-cache reuse, forcing us to re-compute the KV cache after every training step, leading to huge latency and memory overheads. We design qTTT to be intentionally minimal: one prefill, tiny query-only updates, then standard decoding.
>
> Meta-learning could be potentially useful to dynamically choose between different test-time scaling strategies and could be a very interesting future extension on top of our work.
>
> Recalling from our theoretical findings (**Corollary 2.5**), score dilution is a direct result of attention. In order to increase the target-distractor logit margin, moving the query vector by updating the query matrix provides the minimal sufficient fix. We also find that this simple and efficient way provides large practical gains on our empirical settings.
>
> We acknowledge that updating the MLPs could possibly help further adapt the model weights to the given context resulting in downstream improvements but leave this for future work.
>
> **Real-world serving via vLLM and CUDA graph overheads**
>
> We acknowledge the reviewer’s point regarding the complexity of test-time training in production. However, we propose a minimal interference strategy that ensures qTTT remains compatible with existing CUDA graph optimizations (e.g., in vLLM):
>
> - Step 1 (**Prefill**): Perform a standard single prefill to populate the Paged KV cache.
>
> - Step 2 (**Adaptation**): Run the brief qTTT adaptation steps outside the captured CUDA graphs. This produces a temporary, low-rank update, ΔWQ  (operationally implemented as a transient LoRA adapter).
>
> - Step 3 (**Decoding**): Decode using the captured graphs by simply injecting the adapter (WQ +ΔWQ ).
> Since the shapes and kernels remain unchanged, the CUDA graphs remain valid, and the memory behavior matches standard vLLM decoding with only a negligible, bounded cost for the adapter application.
>
> We conducted a comprehensive latency analysis (fully detailed in **Appendix H, Tables 10-11**). Below, we show the concise summary comparing qTTT against "Thinking Tokens" (inference-time compute baseline) and "Best-of-N" (sampling baseline).
>
> | #Input tokens |  Prefill (s)  |    qTTT (s)   | Thinking (s) | Best-of-N (s) |
> |:--------------:|:-------------:|:-------------:|:-------------------:|:-------------:|
> | 32,000         | 58.45 ± 2.34  | 72.11 ± 2.88  | 72.09 ± 2.88        | 68.69 ± 2.75  |
> | 128,000        | 233.81 ± 9.35 | 247.47 ± 9.90 | 247.41 ± 9.90       | 233.81 ± 9.35 |
>
>
> | #Input tokens |   Prefill (s)  |    qTTT (s)    | Thinking (s) |  Best-of-N (s) |
> |:--------------:|:--------------:|:--------------:|:-------------------:|:--------------:|
> | 32,000         | 88.53 ± 3.54   | 109.01 ± 4.36  | 109.00 ± 4.36       | 97.07 ± 3.88   |
> | 128,000        | 354.13 ± 14.17 | 374.61 ± 14.98 | 374.67 ± 14.99      | 354.13 ± 14.1
>
> We find that the wall-clock time for all three test-time compute strategies, qTTT, thinking, and best-of-N, is very close. We also note that prefilling the KV cache dominates most of the decoding time, especially for longer sequence lengths. This motivates our approach of keeping the K/V weights frozen, without which the prefill would need to be recomputed after every training step.

---

### Author Response · Authors · 2025-11-27
**Rebuttal Summary**

We thank the reviewers for their constructive feedback and insightful questions. We are encouraged that reviewers recognized the novelty of our query-only update (Reviewer SSXV, BjzK), praised the compute-aware design and efficiency of the approach (Reviewer SSXV, Yzb3), and acknowledged the strong performance improvements (Reviewer BjzK) and theoretical formalization of the problem (Reviewer Yzb3).

The reviewers’ main concerns centered on:

- Validity: Whether "score dilution" is the true cause of failure compared to other factors (e.g., RoPE, lack of training samples).
- Baselines: FLOP-matched test-time scaling baselines (e.g., Best-of-N, Beam Search).
- Scaling: Whether the method works on larger models (e.g., >8B parameters).
- Efficiency: Practical inference latency and wall-clock time overhead.

We conducted extensive new experiments to address all of these concerns. We highlight the key new observations added to the paper below:
- **Validated "Score Dilution" via Attention Mass (Appendix E, Tables 2 & 3):**
We analyzed attention mass on target tokens (addressing R-SSXV). We found that score dilution persists even when using RoPE and even when sufficient long-context training samples exist. qTTT consistently recovers attention mass on targets where standard attention fails.

- **Added FLOP-Matched Baselines (Appendix G, Tables 8 & 9):** We compared qTTT against Best-of-N and Beam Search with equivalent compute budgets (addressing R-SSXV, R-Yzb3). qTTT significantly outperforms both baselines (e.g., +3.1% vs Best-of-N on LongBench-v2).

- **Scaled to Larger Models (Appendix G, Tables 6 & 7):**
We evaluated Qwen3-32B (addressing R-BjzK). Performance gains hold at scale. qTTT improves Qwen3-32B by ~6% on LongBench-v2 and ~4% on ZeroScrolls compared to thinking baselines.

- **Inference Latency Analysis (Appendix H, Tables 10 & 11):**
We measured wall-clock time including prefill and adaptation (addressing R-BjzK, R-Yzb3). Since prefill dominates total time, the overhead of qTTT is minimal. Total latency is comparable to thinking tokens and best-of-N.

- **Robustness Analysis (Appendix D, Table 1):**
We added a learning rate ablation (addressing R-SSXV). qTTT is robust across a wide range of learning rates (3e-5 to 1e-6).

We believe these new results robustly validate the mechanism and practical utility of qTTT. We kindly invite the reviewers to check the detailed results in the updated paper and reconsider their scores.

---

### Meta-Review · Area_Chair_DZyd · 2026-01-11

**Summary:**

This paper proposes a query-only test-time training (qTTT) method for LLMs to better handle long contexts. Given a prefilled query KV cache, it performs test-time training by updating only the query weights via gradient descent. The experiments demonstrate that the proposed method outperforms existing test-time training methods and baselines.

The reviewers' main concerns can be summarized as:
- Experiments and analysis on efficiency and latency are insufficient.
- Comparison with baseline test-time scaling methods (e.g., BoN) is needed
- Weak motivation (regarding on score dilution) and justification of the method.

During the rebuttal phase, the authors provided additional experiments and analysis, and the concerns were partially addressed. The AC and reviewers acknowledged the merits of the proposed method, and based on the overall review opinions, the AC's decision is to accept this paper.

**Reviewer Concerns:**

The detailed concerns raised by the reviewers are,
- Reviewer BjzK: Insufficient analysis on efficiency. Concerns about whether the method can work in a practical scenario.
- Reviewer Yzb3: Insufficient analysis on inference latency. Comparison with more baseline methods. Questions on low-quality text inputs.
- Reviewer SSXV: Weak motivation and justification on score dilution. Comparison with more baselines. Insufficient ablation study.

**Reviewer Scores:**

- Reviewer BjzK: 6 → 6
- Reviewer Yzb3: 4 → 6, as mentioned in the discussion.
- Reviewer SSXV: 2 → 4 or 6, as the provided analysis of score dilution and ablation was not fully satisfactory. However, the AC believes that if the discussion had continued, considering the authors' responses, there is a high likelihood that the score would have increased from 4 to 6.

---

### Decision · Program_Chairs · 2026-01-26

Accept (Poster)